# Re-evaluation of neuronal P2X7 expression using novel mouse models and a P2X7-specific nanobody

Karina Kaczmarek-Hajek[1†], Jiong Zhang[1,2†‡], Robin Kopp[2†], Antje Grosche[3,4], Björn Rissiek[5], Anika Saul[1§], Santina Bruzzone[6], Tobias Engel[7], Tina Jooss[2], Anna Krautloher[2], Stefanie Schuster[8], Tim Magnus[5], Christine Stadelmann[9], Swetlana Sirko[4,10], Friedrich Koch-Nolte[11], Volker Eulenburg[8,12], Annette Nicke[1,2]*

[1]Department of Molecular Biology of Neuronal Signals, Max Planck Institute for Experimental Medicine, Göttingen, Germany; [2]Walther Straub Institute for Pharmacology and Toxicology, Ludwig-Maximilians-Universität München, Munich, Germany; [3]Institute for Human Genetics, University of Regensburg, Regensburg, Germany; [4]Department of Physiological Genomics, Ludwig-Maximilians-Universität München, München, Germany; [5]Department of Neurology, University Hospital Hamburg-Eppendorf, Hamburg, Germany; [6]Department of Experimental Medicine and CEBR, University of Genova, Genova, Italy; [7]Department of Physiology and Medical Physics, Royal College of Surgeons in Ireland, Dublin, Ireland; [8]Institute of Biochemistry, University Erlangen-Nürnberg, Erlangen, Germany; [9]Institute of Neuropathology, University Medical Center, Göttingen, Germany; [10]Institute of Stem Cell Research, Helmholtz Center Munich, German Research Center for Environmental Health (GmbH), Neuherberg, Germany; [11]Department of Immunology, University Hospital Hamburg-Eppendorf, Hamburg, Germany; [12]Department of Anaesthesiology and Intensive Care Therapy, University of Leipzig, Leipzig, Germany

*For correspondence:
annette.nicke@lrz.uni-muenchen.
de

[†]These authors contributed equally to this work

Present address: [‡]Department of Neurology, University Medical Center, Göttingen, Germany; [§]Synaptic Systems GmbH, Göttingen, Germany

**Abstract** The P2X7 channel is involved in the pathogenesis of various CNS diseases. An increasing number of studies suggest its presence in neurons where its putative functions remain controversial for more than a decade. To resolve this issue and to provide a model for analysis of P2X7 functions, we generated P2X7 BAC transgenic mice that allow visualization of functional EGFP-tagged P2X7 receptors *in vivo*. Extensive characterization of these mice revealed dominant P2X7-EGFP protein expression in microglia, Bergmann glia, and oligodendrocytes, but not in neurons. These findings were further validated by microglia- and oligodendrocyte-specific P2X7 deletion and a novel P2X7-specific nanobody. In addition to the first quantitative analysis of P2X7 protein expression in the CNS, we show potential consequences of its overexpression in ischemic retina and post-traumatic cerebral cortex grey matter. This novel mouse model overcomes previous limitations in P2X7 research and will help to determine its physiological roles and contribution to diseases.
DOI: https://doi.org/10.7554/eLife.36217.001

## Introduction

The P2X7 receptor differs from all other P2X family members by its low sensitivity to ATP, a particularly long intracellular C-terminus, and its ability to trigger various short and long-term cellular events

**eLife digest** The human body relies on a molecule called ATP as an energy source and as a messenger. When cells die, for example if they are damaged or because of inflammation, they release large amounts of ATP into their environment. Their neighbors can detect the outpouring of ATP through specific receptors, the proteins that sit at the cell's surface and can bind external agents.

Scientists believe that one of these ATP-binding receptors, P2X7, responds to high levels of ATP by triggering a cascade of reactions that results in inflammation and cell death. P2X7 also seems to play a role in several brain diseases such as epilepsia and Alzheimer's, but the exact mechanisms are not known. In particular, how this receptor is involved in the death of neurons is unclear, and researchers still debate whether P2X7 is present in neurons and in other types of brain cells.

To answer this, Kaczmarek-Hájek, Zhang, Kopp et al. created genetically modified mice in which the P2X7 receptors carry a fluorescent dye. Powerful microscopes can pick up the light signal from the dye and help to reveal which cells have the receptors. These experiments show that neurons do not carry the protein; instead, P2X7 is present in certain brain cells that keep the neurons healthy. For example, it is found in the immune cells that 'clean up' the organ, and the cells that support and insulate neurons. Kaczmarek-Hájek et al. further provide preliminary data suggesting that, under certain conditions, if too many P2X7 receptors are present in these cells neuronal damage might be increased. It is therefore possible that the brain cells that carry P2X7 indirectly contribute to the death of neurons when large amounts of ATP are released.

The genetically engineered mouse designed for the experiments could be used in further studies to dissect the role that P2X7 plays in diseases of the nervous system. In particular, this mouse model might help to understand whether the receptor could become a drug target for neurodegenerative conditions.

DOI: https://doi.org/10.7554/eLife.36217.002

like the induction of dye uptake and cell death (*Saul et al., 2013*; *Surprenant et al., 1996*; *Li et al., 2015*; *Harkat et al., 2017*; *Di Virgilio et al., 2018*), which are incompletely understood and have mostly been described in cell culture systems. Probably best studied is its central role in NLRP3 (NOD-like receptor family, pyrin domain containing 3) inflammasome activation, cytokine maturation, and inflammation that was originally established in P2X7 knockout ($P2rx7^{-/-}$) mouse models (*Chessell et al., 2005*; *Solle et al., 2001*; *Di Virgilio et al., 2017*). A growing body of evidence suggests that an increased P2X7 receptor function plays a role in various diseases of the central nervous system (CNS) and peripheral nervous system (PNS) and further supports its importance as a drug target (*Bhattacharya and Biber, 2016*; *Rassendren and Audinat, 2016*; *Sperlágh and Illes, 2014*; *Sociali et al., 2016*).

P2X7 receptors are expressed in cells of hematopoietic origin as well as different types of glial, epithelial, and endothelial cells. The presence and function of P2X7 receptors in neurons, however, remains a matter of debate (*Illes et al., 2017*; *Miras-Portugal et al., 2017*), although an increasing amount of literature describing neuronal P2X7 functions imply a wide acceptance of this view (e.g. in [*Sperlágh and Illes, 2014*; *Brown et al., 2016*; *Engel et al., 2012*; *Gulbransen et al., 2012*; *Jimenez-Pacheco et al., 2013*]). In support of a neuronal expression, *P2rx7* mRNA was detected in neurons (*Yu et al., 2008*), P2X7 protein was identified in neuronal cell culture (*Ohishi et al., 2016*), and P2X7 receptors were pharmacologically shown to facilitate postsynaptic efficacy and affect neurotransmitter release (reviewed in [*Sperlágh and Illes, 2014*]). However, mRNA expression might not necessarily correlate with synthesis of the respective protein (*Carpenter et al., 2014*), selectivity of the available P2X7-specific antibodies has been questioned (*Anderson and Nedergaard, 2006*; *Sim et al., 2004*), and pharmacology of purinergic receptors is rather complex (*Anderson and Nedergaard, 2006*; *Compan et al., 2012*; *Nörenberg et al., 2016*). Also, it has been difficult to differentiate between direct effects of neuronal P2X7 activation and indirect effects of ATP-activated neurotransmitter release from glia cells (*Sperlágh and Illes, 2014*; *Illes et al., 2017*; *Miras-Portugal et al., 2017*).

Taken together, the scarcity of information regarding the localization and the molecular and physiological functions of P2X7 receptors in the nervous system stands in sharp contrast to its proposed role as a drug target. To conclusively resolve these important questions, we generated transgenic mouse lines that overexpress EGFP-tagged P2X7 under the control of a BAC-derived mouse P2X7 gene (*P2rx7*) promoter. These mice allow the direct and indirect visualization of P2X7, its purification, and determination of functional consequences of its overexpression. Using this model, we provide the first comprehensive and quantitative analysis of the distribution of P2X7 protein within the CNS.

## Results

### Generation of P2X7-EGFP BAC transgenic mice

The *P2rx7* cDNA was obtained from C57BL/6 mouse brain and C-terminally fused to the EGFP-sequence via a Strep-tagII-Gly-7xHis-Gly linker sequence (*Figure 1—figure supplement 1A*) to provide additional labeling/purification options and minimize interference with the receptor function. As two allelic P2X7 variants, 451P ('wt') and 451L (SNP, present in C57BL/6), with different functionality have been described (*Adriouch et al., 2002*; *Sorge et al., 2012*), the 'wt' L451P-variant was also generated by site directed mutagenesis. Efficient expression and functionality of the full-length proteins were confirmed by SDS-PAGE, patch-clamp analysis, and ATP-induced ethidium uptake in HEK cells (*Figure 1—figure supplement 1B–E*). Both variants and the non-tagged receptors revealed similar $EC_{50}$ values, indicating that the dye uptake properties of the P2X7 receptor were not influenced by the EGFP-tag. Also, current kinetics were virtually identical. Next, BAC clone RP24-114E20, containing the full length *P2rx7* and more than 100 kb of the 5′region was modified accordingly by insertion of the Strep-His-EGFP sequence in exon 13 to preserve the exon-intron structure of the gene (*Figure 1A*). Upon verification by Southern blotting (*Figure 1B*, *Figure 1—figure supplement 1F*) and sequencing, the linearized BAC was injected into pronuclei of FVB/N mouse oocytes (451L background). In total, 4 (451L) and 10 (451P) germline transmitters were obtained and five lines (451L: lines 46, 59 and 61; 451P: lines 15 and 17) were selected for initial characterization as described below (*Figure 1C* and *Figure 2—figure supplement 1*). Subsequent experiments were performed with the highest expressing line 17.

### Expression, membrane targeting, and function of P2X7-EGFP in transgenic mice

Southern blot analysis revealed integration of 4–15 BAC copies in the different lines. The copy numbers correlated well with the respective P2X7-EGFP protein expression levels (*Figure 1C*), suggesting the functionality of most if not all integrated *P2rx7* BAC transgenes. Endogenous P2X7 protein synthesis was unaffected by the P2X7-EGFP overexpression (*Figure 1—figure supplement 1G*). Purification of P2X7-EGFP protein via Ni-NTA agarose demonstrated co-purification of endogenous P2X7 subunits confirming efficient co-assembly of tagged and non-tagged subunits (*Figure 1D*). In agreement with correct plasma membrane targeting, deglycosylation with endoglycosidase H and PNGase F revealed efficient complex glycosylation, indicating that the EGFP-tag did not disturb folding and ER-exit of the transgenic P2X7-EGFP protein (*Figure 1E*). To demonstrate functionality of the overexpressed P2X7-EGFP protein, the transgenic mice were, upon backcrossing into C57BL/6 for 8–10 generations, mated to *P2rx7*$^{-/-}$ mice (in C57BL/6) to obtain 'rescue' mice (*Table 1*) that express only the transgenic but not the endogenous P2X7 (*Figure 1F*). FACS analysis of microglia from these mice confirmed that the transgene is able to fully rescue the ATP-induced DAPI uptake, which is absent in these cells from *P2rx7*$^{-/-}$ mice (*Figure 1G*). Comparison of the kinetic and efficiency of DAPI uptake by simultaneous analysis of pooled and differentially labeled microglia revealed a stronger increase in the rescue mice compared to wt mice, most likely due to a higher number of functional P2X7 receptors at the cell surface. The specificity of the DAPI uptake was demonstrated using the P2X7 antagonist A438079 (*Figure 1—figure supplement 2*).

### Analysis of P2X7-EGFP localization in the brain

To determine the overall pattern of P2X7 localization in the brain, 3, 3′-Diaminobenzidine (DAB) staining with EGFP-specific antibodies was performed on brain slices from all mouse lines. As shown

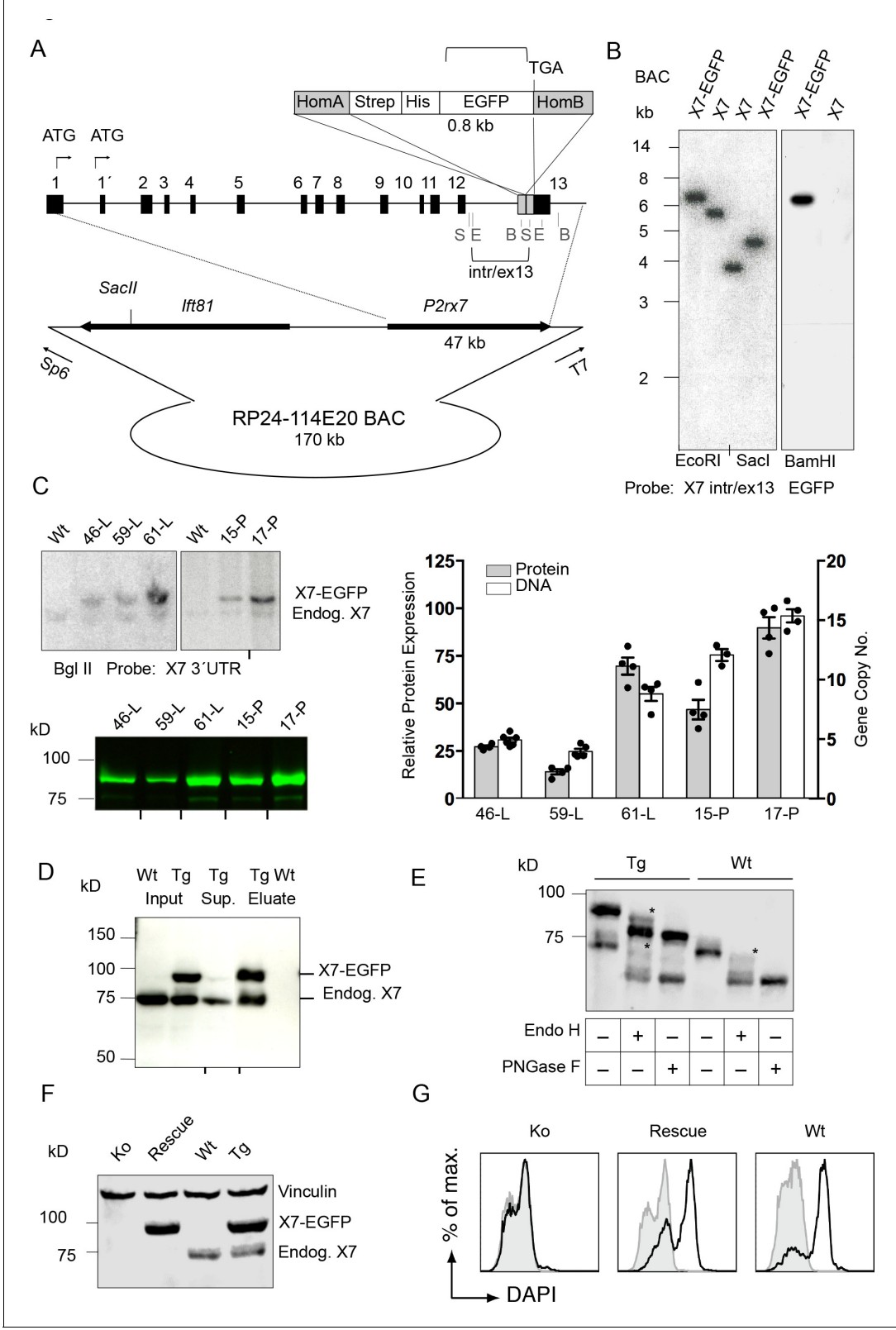

**Figure 1.** Generation and validation of BAC transgenic P2X7-EGFP mice. (**A**) Scheme of the BAC clone containing the full-length *P2rx7* plus about 103 kb (5′) and 10 kb (3′) flanking sequences. A Strep-His-EGFP cassette (0.8 kb) flanked by two homology arms (grey boxes) was inserted directly upstream of the stop codon into exon 13 of the *P2rx7*. Before pronuclear microinjection, the BAC-P2X7-EGFP construct was linearized at a unique SacII site. B, E, and S indicate BamHI, EcoRI, and SacI restriction sites, respectively. The probes used for Southern blot analysis are indicated as brackets. (**B**) Southern

*Figure 1 continued on next page*

*Figure 1 continued*

blot analysis of BAC DNA using these probes (X7 intr/ex13, EGFP) confirmed homologous recombination and correct integration of the Strep-His-EGFP cassette into the BAC. (C) Comparison of copy number and protein expression in different BAC transgenic P2X7-EGFP lines. Representative SDS-PAGE (direct EGFP fluorescence, 60 μg total protein/lane) and Southern blot data are shown (transgenic *P2rx7*, 5277 bp; endogenous *P2rx7*, 4561 bp). Black marks at the bottom indicate where replicates were excised from the figures. Data from 3–6 individual mice are represented. (D) Co-purification of endogenous P2X7 subunits with transgenic receptors. Protein complexes were purified under non-denaturing conditions via Ni-NTA agarose from dodecylmaltoside (0.5%) brain extracts (line 59). A representative result of n > 5 experiments with different lines is shown. P2X7-specific antibody: Synaptic Systems (E) Deglycosylation analysis of endogenous and transgenic P2X7. Protein extracts from spinal cord of wt and line 17 transgenic mice were treated with endoglycosidases as indicated and P2X7 protein was detected by immunoblotting (P2X7-specific antibody, Synaptic Systems). Asterisks indicate Endo H-resistant complex glycosylated protein. A representative result of n > 5 experiments with different organs is shown. (F) P2X7-EGFP (line 17) was crossed into *P2rx7$^{-/-}$* background. Western blot analysis with an P2X7-specific antibody (Synaptic Systems) confirmed successful deletion of the endogenous P2X7 in this rescue mouse. (G) FACS analysis of microglia showing rescue of ATP-induced (1 mM) DAPI uptake by the P2X7 rescue (line 59) microglia in comparison to wt and *P2rx7$^{-/-}$* microglia. A representative result from n = 3 animals is shown.

DOI: https://doi.org/10.7554/eLife.36217.003

The following figure supplements are available for figure 1:

**Figure supplement 1.** Expression and functionality of the P2X7-EGFP constructs in HEK cells and expression of the transgene in mice.
DOI: https://doi.org/10.7554/eLife.36217.004

**Figure supplement 2.** Efficiency and specificity of ATP-induced DAPI uptake in primary brain microglia from wt and P2X7-EGFP transgenic mice.
DOI: https://doi.org/10.7554/eLife.36217.005

for five selected lines (*Figure 2A*, *Figure 2—figure supplement 1*), specific labeling with identical patterns (*Table 2*) was obtained. A particularly high P2X7-EGFP density was found in the molecular layers of the cerebellar cortex. In addition, strong labeling was detected in the molecular layer of the dentate gyrus (DG), the cerebral cortex and olfactory bulb as well as the thalamus, hypothalamus, substantia nigra, and ventral pons. Comparison of endogenous and transgenic P2X7 levels in different brain regions (*Figure 2B*) showed similar protein ratios and tissue-specific intensities, demonstrating that expression of the transgene mirrored both the expression pattern and expression level of endogenous P2X7 and thus implying that important regulatory elements governing P2X7 expression are preserved and functional in the chosen BAC construct. Due to the dense but diffuse signal and a higher background fluorescence (probably due to structural organization and/or a high content of endogenous fluorophores) in the cerebellar cortex, identification of cellular structures and P2X7-EGFP-expressing cell types proved difficult in adult cerebellum (*Figure 2—figure supplement 2A–C*): Using confocal microscopy, no conclusive co-localization was seen with Purkinje cell

**Table 1.** Mice

| Strain | Official name | Origin |
|---|---|---|
| P2X7-EGFP | FVB/N-Tg(RP24-114E20P2X7-StrepHis-EGFP)Ani Lines 46, 59 (also in BL/6N), 61 | This study |
| P2X7$^{451P}$-EGFP | FVB/N-Tg(RP24-114E20P2X7$^{451P}$-StrepHis-EGFP)Ani Lines 15, 17 (also in BL/6N) Transgenes were backcrossed into C57BL/6 for at least eight generations | This study |
| *P2rx7$^{fl/fl}$* | B6-P2rx7$^{tm1c(EUCOMM)Wtsi}$ | This study (B6-P2rx7$^{tm1a(EUCOMM)Wtsi}$ x FLPe deleter mouse Gt(ROSA) 26Sor$^{tm1(FLP1)Dym}$ [**Farley et al., 2000**]) |
| *P2rx7$^{-/-}$* | B6-P2rx7$^{tm1d(EUCOMM)Wtsi}$ | This study (P2rx7$^{fl/fl}$ x EIIa-Cre mouse Tg(EIIa-cre)C5379Lmgd [**Lakso et al., 1996**]) |
| P2X7 rescue | B6-P2rx7$^{tm1d(EUCOMM)Wtsi}$//B6.Cg-Tg(RP24-114E20P2X7-StrepHis-EGFP)Ani | This study (P2rx7$^{-/-}$ x P2X7-EGFP line 59 and 17 in C57BL/6) |
| Microglia-specific P2X7 knock-out | B6-P2rx7$^{tm1c(EUCOMM)Wtsi}$//B6-Cx3cr1$^{tm1.1(cre)Jung}$ | This study (P2rx7$^{fl/fl}$ x Cx3cr1$^{tm1.1(cre)Jung}$ [**Yona et al., 2013**]) |
| Oligodendrocyte-specific P2X7 knock-out | B6-P2rx7$^{tm1c(EUCOMM)Wtsi}$//B6- Cnp$^{tm1(cre)Kan}$ | This study (P2rx7$^{fl/fl}$ x CNP-Cre mouse Cnp$^{tm1(cre)Kan}$ [**Lappe-Siefke et al., 2003**]) |

If not otherwise noted, mice of both genders (9–14 weeks) in FVB/N background were used. Given that FVB/NJ mice are homozygous for the retinal degeneration 1 allele of Pde6b$^{rd1}$, retinal stainings were performed in C57b/6 or FVB/N/C57b/6 hybrid mice.

DOI: https://doi.org/10.7554/eLife.36217.006

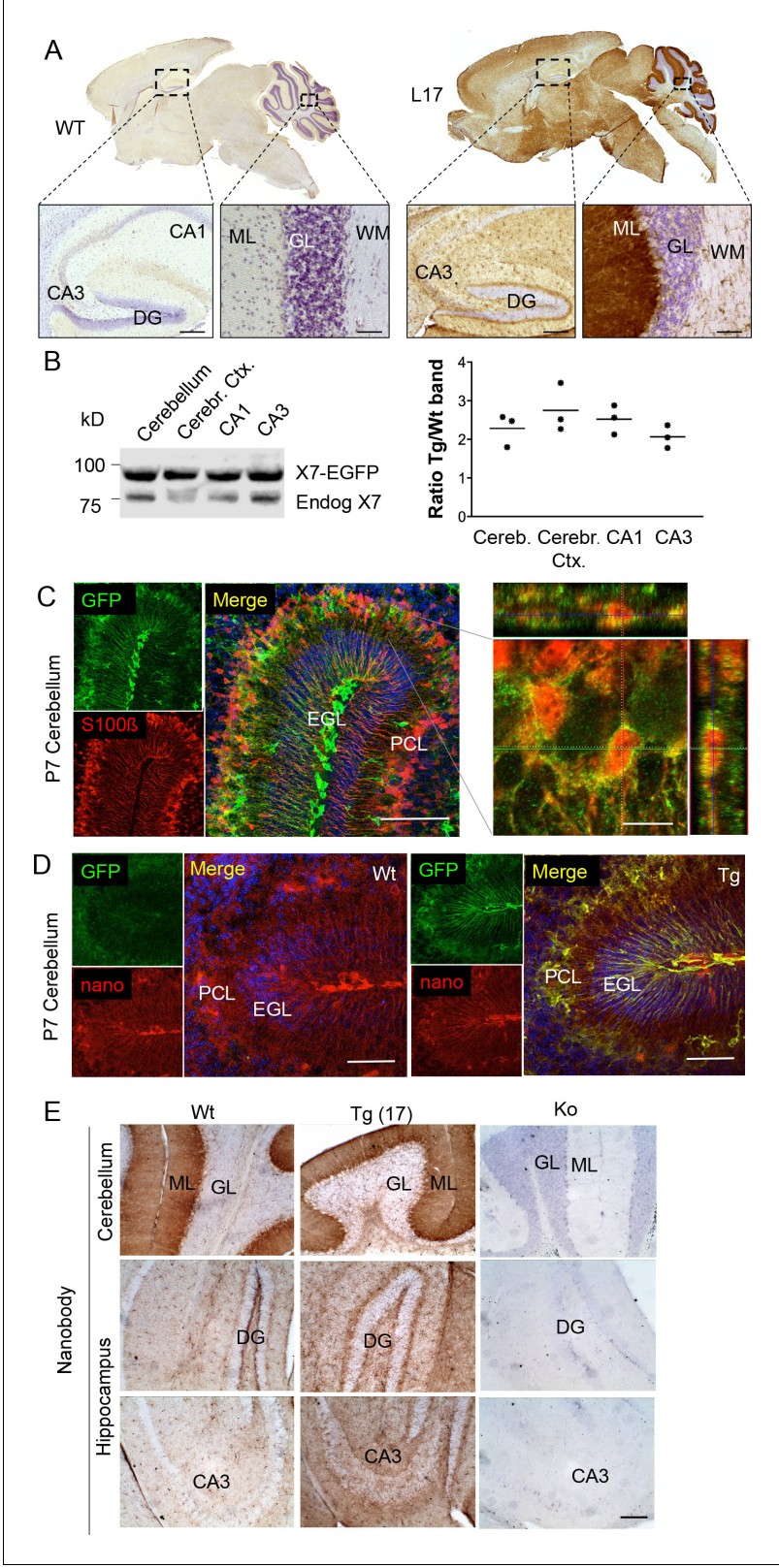

**Figure 2.** Distribution pattern of transgenic P2X7-EGFP. (**A**) DAB staining with an antibody against GFP (A11122, Thermo Fisher Scientific). Scale bars: 200 μm and 50 μm in hippocampus and cerebellum, respectively. A representative result from at least three animals is shown (**B**) Ratios of transgenic (line 17) and endogenous P2X7 protein in different brain regions. Protein extracts (1% NP40) were prepared and 75 μg per lane separated by SDS-
*Figure 2 continued on next page*

*Figure 2 continued*

PAGE. Bands were quantified upon western blotting by infrared imaging with antibodies against P2X7 (Synaptic Systems) and fluorescent secondary antibodies (LI-COR 680RD dk anti rb). Data are presented as means from three animals. (C) Co-labeling of line 17 P7 cerebellum with antibodies against GFP (A10262, Thermo Fisher Scientific) and S100β (S2532, Sigma Aldrich). A typical staining pattern for radial glia is seen. The close up of a representative area in the Purkinje cell layer (right) shows punctate P2X7 staining on cells with Bergmann glia morphology. Cell nuclei were counterstained with DAPI (blue). Scale bars represent 100 μm and 10 μm, respectively. CA1/3, cornu ammonis regions 1/3; DG, dentate gyrus; ML, molecular layer; GL, granular layer; WM, white matter; EGL, external granular layer; PCL, Purkinje cell layer. (D) Co-labeling of wt and tg line 17 cerebellar slices from P7 pubs with an anti-GFP antibody (A10262, Thermo Fischer Sci.) and the novel P2X7-specific nanobody-rbIgG fusion construct 7E2-rbIgG (*Danquah et al., 2016*) confirms the endogenous expression pattern and the specificity of the P2X7-EGFP signal. Representative results from n = 3 (Tg) and n = 2 (Wt) pubs are shown. Scale bar: 50 μm, DAPI staining in blue. PCL, Purkinje cell layer; GL, granular layer; ML, molecular layer; DG, dentate gyrus; CA3, cornu ammonis region 3; EGL, external granular layer. (E) Comparison of DAB staining in transgenic P2X7-EGFP mice, wt, and P2X7$^{-/-}$ mice with 7E2-rbIgG (*Danquah et al., 2016*). Scale bar: 100 μm. Representative results from three animals per line (line 17 and wt) are shown. For antibodies not specified in the legend see Key resources table.

DOI: https://doi.org/10.7554/eLife.36217.007

The following figure supplements are available for figure 2:

**Figure supplement 1.** Identical expression patterns in five transgenic lines and wt animals.
DOI: https://doi.org/10.7554/eLife.36217.008

**Figure supplement 2.** P2X7-EGFP immunofluorescence in the cerebellum.
DOI: https://doi.org/10.7554/eLife.36217.009

(calbindin D28k) and synaptic (vGlut2) marker proteins nor with astrocytes/Bergmann glia (GFAP or S100β). However, analysis of the cerebellum from animals at postnatal day 7 (P7), before Bergmann glia microdomains are formed (*Grosche et al., 2002*), revealed a more structured and clearer GFP signal that aligned with the cell bodies and radial extensions of S100β-immunopositive cells with typical Bergman glia morphology (*Figure 2C*). Thus, we conclude that P2X7 is expressed in Bergmann glia, in agreement with previous findings (*Habbas et al., 2011*). However, due to the localization in microdomains that give a very diffuse pattern, co-localization with the intracellular Bergmann glia marker GFAP, which visualizes mainly their radial extensions, could not be detected and co-

**Table 2.** P2X7-EGFP protein expression in comparison with other P2X7 (reporter) mouse models (Tg (P2rx7 EGFP)FY174Gsat, www.gensat.org, P2rx7$^{hP2RX7}$ (*Metzger et al., 2017*)

| | Transgenic P2X7-EGFP | P2X7 reporter (Gensat) | Humanized P2X7 |
|---|---|---|---|
| Brain region | Fusion protein | Soluble EGFP | RNA |
| Hippocampus | M $_{(ML+)}$, O, A | N, G | N (CA3+), O, A |
| Cerebral Cort. | M, O, A | N, G | M, O, A |
| Midbrain | M, O | N, G | nd |
| Thalamus | M, O | G | nd |
| Hypothalamus | M, O | N, G | nd |
| Cerebellum | M, O, A, BG (ML+) | N, G, BG (ML+) | O, A |
| Olfactory bulb | M | G | nd |
| Ventricle | EC | nd | nd |
| Corpus callosum | M, O | nd | nd |

A high density of positive cells in a specific area is indicated by +whereas the presence in specific cells or structures is indicated by letters (N, neuron; PC, Purkinje cell; A, astrocyte; M, microglia; O, oligodendrocyte; BG, Bergmann glia; G, glia-like; EC, ependymal cells, ML, molecular layer; GCL, granular cell layer; nd, not determined). Data from the Gensat mouse are according to information given on the gensat web site (www.gensat.org) for fluorescence images listed under confirmed expression veracity.

DOI: https://doi.org/10.7554/eLife.36217.010

localization with S100β was only dissolved during postnatal development. In addition to Bergman glia, P2X7 is present in microglia of the cerebellum, which was confirmed in acutely dissociated cells from adult tissue (*Figure 2-figure supplement 2D*). Based on our data (*Figure 2E*, *Figure 2—figure supplement 2B*), we exclude the expression in Purkinje cells in both adult and P7 mice. Specificity of the GFP labeling and congruency with the endogenous P2X7 expression pattern was further confirmed using a novel mouse P2X7-specific heavy chain antibody (nanobody 7E2 fused to the hinge, CH2 and CH3 domains of rabbit-IgG (7E2-rbIgG), see (*Danquah et al., 2016*) and Materials and methods) (*Figure 2D and E*).

Co-immunolabeling with cell type-specific markers (*Figure 3A–F*) and quantification of GFP-positive cells in the CA1 region of the hippocampus (*Figure 3G*) demonstrated that P2X7-EGFP is predominantly (57 ± 14%) expressed in microglial cells (93% of all Iba1-positive cells), while the majority, if not all, of the remaining GFP-positive cells (47 ± 10%) belong to the oligodendroglial lineage and co-express Olig2 (87% of Olig2-positive and 95% of NG2-positive oligodendrocyte precursor cells). In addition, 7 ± 4% of P2X7-EGFP-expressing cells represent S100β-positive but GFAP-negative cells. These cells comprised 8% of all S100β-positive cells and may represent either GFAP-negative astroglial cells or oligodendrocyte precursor cells. This distribution is in agreement with functional findings (*Jabs et al., 2007*) and cell type-specific RNA sequencing data (*P2rx7* mRNA in microglia/oligodendrocytes/astrocytes ≈ 28/26/5 fragments per kilobase of transcript sequence per million mapped fragments) obtained from cerebral cortex (*Zhang et al., 2014*) (http://web.stanford.edu/group/barres_lab/brain_rnaseq.html). Likewise, co-staining with Sox9 (for astrocytes) or neuronal (NeuN, MAP2) and synaptic (VGlut1, PSD95, synaptophysin, VGAT) markers did not reveal any overlap in the CA1, CA3, and dentate gyrus (*Figure 3A*, *Figure 3—figure supplements 1*, *2* and *3B*). A clear band of more intense GFP signal is regularly detected in the molecular layer of the dentate gyrus (e.g. *Figure 2A*, *Figure 3—figure supplement 1B*, bottom-right panel) and was attributed to a higher number and/or more ramified morphology of microglia that align at the border of the granular layer. In support of this explanation, the thickness of the band in this region equals the radius of microglia with their extensions (*Figure 3—figure supplement 3A*). As P2X7 protein expression has been described in nestin-positive neuronal/glia precursor cells in the subgranular zone (*Rozmer et al., 2017*), we also performed co-labeling of EGFP with nestin in this region but did not detect any co-localization (*Figure 3—figure supplement 4*). Likewise, no co-staining of EGFP with neurons was seen in other regions with a strong EGFP signal like basal ganglia, hypothalamus, and pons (*Figure 3—figure supplement 5*).

In addition, we performed co-stainings of brain sections with the commercially available P2X7-specific antibodies and the nanobody-rbIgG fusion construct 7E2-rbIgG (*Danquah et al., 2016*). However, the commercially available antibodies yielded unspecific or insufficient staining (either in comparison to *P2rx7*[-/-] mice or in the P2X7-EGFP overexpressing line 17) (*Figure 3—figure supplement 6*). In contrast, 7E2-rbIgG showed specific staining of endogenous P2X7 protein in wild-type (wt) but not *P2rx7*[-/-] mice (*Figure 2E*) and clear overlap with the transgenic P2X7-EGFP (*Figure 2E*, *Figure 3—figure supplement 6*). To further verify that the observed transgene expression pattern correlates with the endogenous P2X7 expression, mice deficient in microglial or oligodendroglial P2X7 were generated by mating *P2rx7*[fl/fl] mice with Cx3cr1[tm1.1(cre)Jung] (*Yona et al., 2013*) and Cnp[tm1(cre)Kan] lines (*Lappe-Siefke et al., 2003*), respectively (specificity of Cre expression is shown in *Figure 3—figure supplement 7*). In comparison to *P2rx7*[fl/fl] mice, Cx3cr1-Cre-positive mice showed 51.5% (±4.5%) and Cnp-Cre-positive mice showed 60.4% (±2.9%) reduction of P2X7 protein in the brain, which correlates well with the percentage of P2X7 expressing cells determined in the brains of our transgenic mice (*Figure 3H*).

## Analysis of P2X7-EGFP localization in other neuronal preparations

Since neuronal P2X7 expression and function has been described in amacrine cells (interneurons) as well as in ganglion cells, photoreceptors, and pigment epithelial cells of the retina (*Sanderson et al., 2014*), we further probed if neuronal P2X7 expression was detectable in this tissue with histologically clear architecture. In contrast to previous reports, however, P2X7-EGFP was exclusively expressed in microglia (*Figure 4A*). Likewise, P2X7-EGFP expression was not found in neurons of the spinal cord, DRG, or of teased sciatic nerve fibers (*Figure 4B–D*, *Figure 4—figure supplement 1*). In Schwann cells of the sciatic nerve fibers, however, P2X7-EGFP was localized to nodes of Ranvier and Schmidt-Lantermann incisures, in perfect agreement with the subcellular distribution pattern of endogenous

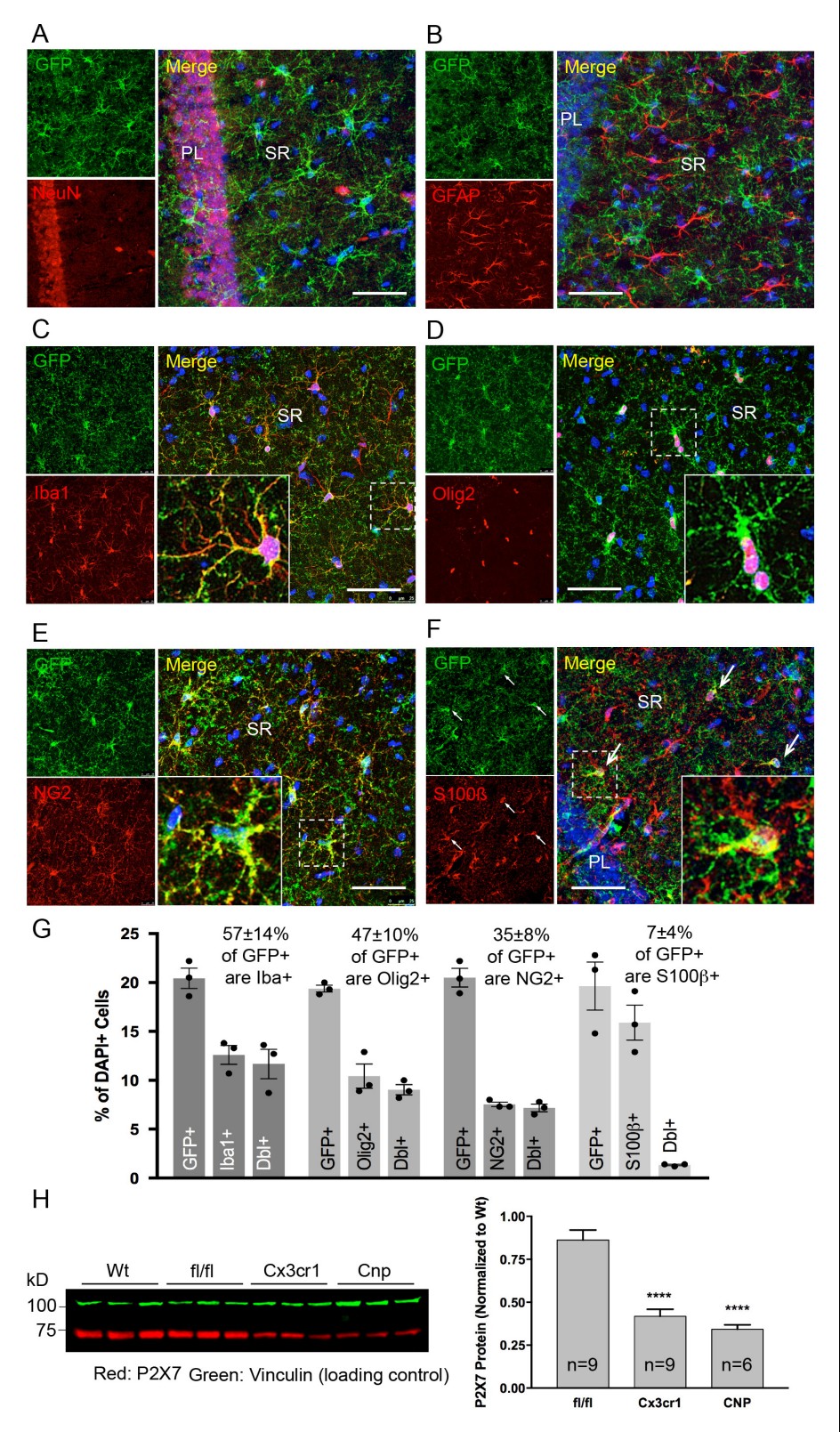

**Figure 3.** Identity and quantity of P2X7-EGFP expressing cell types in the CA1 region and comparison with P2X7 expression in wt mice. (A–F) Co-labeling of tg line 17 brain slices with anti-GFP antibody (ab6556, Abcam; A10262, Thermo Fisher Scientific) and antibodies for the indicated marker proteins (GFAP (MAB360, Millipore), S100β (S2532, Sigma Aldrich)). Hippocampal CA1 regions are shown. Arrows indicate co-staining for S100β and GFP. Cell

*Figure 3 continued on next page*

*Figure 3 continued*

nuclei were counterstained with DAPI (blue). PL, pyramidal cell layer; SR, stratum radiatum. Scale bar: 50 µm (G) Quantitative analysis of 10 'counting boxes' (as shown in C–F) from five sections/mouse in each experiment. Bars represent mean ±SEM of three independent experiments/animals (total cell numbers in transgenic versus wt animals were: 14.4% vs. 12.2% Iba1 +cells, 10.4% vs. 11.0% Olig2 +cells, 7.3% vs. 8.4% NG2 +cells, 16.1 vs. 14.1% S100β + cells). (H) Quantitative analysis of P2X7 protein reduction in conditional P2X7$^{-/-}$ mice (CNP-cre, Cx3cr1-cre). 75 µg cerebrum extracts (1% NP40) were analyzed by western blotting and infrared imaging with antibodies against P2X7 (Synaptic Systems) and fluorescent secondary antibodies (LI-COR 680RD dk anti-rb; LI-COR 800CW gt anti-ms). Data were normalized to P2X7 protein in wt animals. Bars represent mean ± SEM from 6 to 9 animals analyzed in three independent experiments. Significance between means was analyzed using two-tailed unpaired Student's *t*-test and indicated as ****p<0.0001 compared to *P2rx7*$^{fl/fl}$. For antibodies not specified in the legend see Key resources table.

DOI: https://doi.org/10.7554/eLife.36217.011

The following figure supplements are available for figure 3:

**Figure supplement 1.** No co-localization of P2X7-EGFP with neuronal/synaptic markers in the CA1 region
DOI: https://doi.org/10.7554/eLife.36217.012

**Figure supplement 2.** No co-localization of P2X7-EGFP with neuronal/synaptic markers in the CA3 region.
DOI: https://doi.org/10.7554/eLife.36217.013

**Figure supplement 3.** Further analysis of P2X7-EGFP expressing cells in the dentate gyrus and CA1 region.
DOI: https://doi.org/10.7554/eLife.36217.014

**Figure supplement 4.** Co-stainings of EGFP (ab6556, Abcam) and the neuronal/astroglial precursor marker nestin in the subgranular zone of the dentate gyrus.
DOI: https://doi.org/10.7554/eLife.36217.015

**Figure supplement 5.** Co-stainings of EGFP (A10262, Thermo Fischer Scientific; ab6556, Abcam) and neuronal markers tyrosine hydroxylase (dopaminergic neurons, (B) and NeuN (C) in the substantia nigra (SN), hypothalamus (Hy) and pons (P).
DOI: https://doi.org/10.7554/eLife.36217.016

**Figure supplement 6.** Comparison of the specificity of commercially available anti-P2X7 antibodies and an anti-P2X7 nanobody-rbIgG heavy chain antibody (7E2-rbIgG) in CA1 (A) and cerebellar (B) slices of adult line 17 mice and P2X7$^{-/-}$ mice.
DOI: https://doi.org/10.7554/eLife.36217.017

**Figure supplement 7.** Cell type-specific Cre-expression in the hippocampal CA1 region and cerebral cortex (Ctx) of Cx3cr1- and CNP-Cre mice.
DOI: https://doi.org/10.7554/eLife.36217.018

P2X7 (*Figure 4C*). At the neuromuscular junction, P2X7-EGFP was found in close association with terminal Schwann cells (S100β-positive), but did not co-localize with them or with the post- (α-bungarotoxin-positive) or presynaptic (synaptophysin-positive) membrane, in contrast to previous findings (*Deuchars et al., 2001*). Based on the localization and morphology, we suggest its presence on kranocytes, a fibroblast-like cell type (*Court et al., 2008*). In agreement with previous reports on P2X7 localization in DRGs (*Zhang et al., 2005*; *Jager and Vaegter, 2016*), we identified P2X7-EGFP in cells that show the localization and typical morphology of satellite glia cells which ensheath large sensory neurons. This was confirmed by co-labeling experiments using the satellite cell marker glutamine synthetase. Unlike in sciatic nerves, however, it was not found in myelin protein zero (MPZ)-positive Schwann cells of the DRG (*Figure 4—figure supplement 1A*). Finally, P2X7-EGFP localization was investigated in the myenteric plexus of the colon, a part of the enteric nervous system, but was also not detected in neurons or GFAP-positive glia (*Figure 4—figure supplement 1B*).

## Consequences of P2X7 overexpression under physiological and pathological conditions

Detailed analyses of brain parenchyma and other types of nervous tissues indicates that the BAC transgenic P2X7-EGFP is correctly regulated in our mouse model and that P2X7 protein is either below detection limit or not synthesized in neurons, at least under physiological conditions in adult mice.

Despite the well-documented role of P2X7 in inflammation and cell death, P2X7 overexpression did not result in an obvious pathology or behavioral changes under physiological conditions

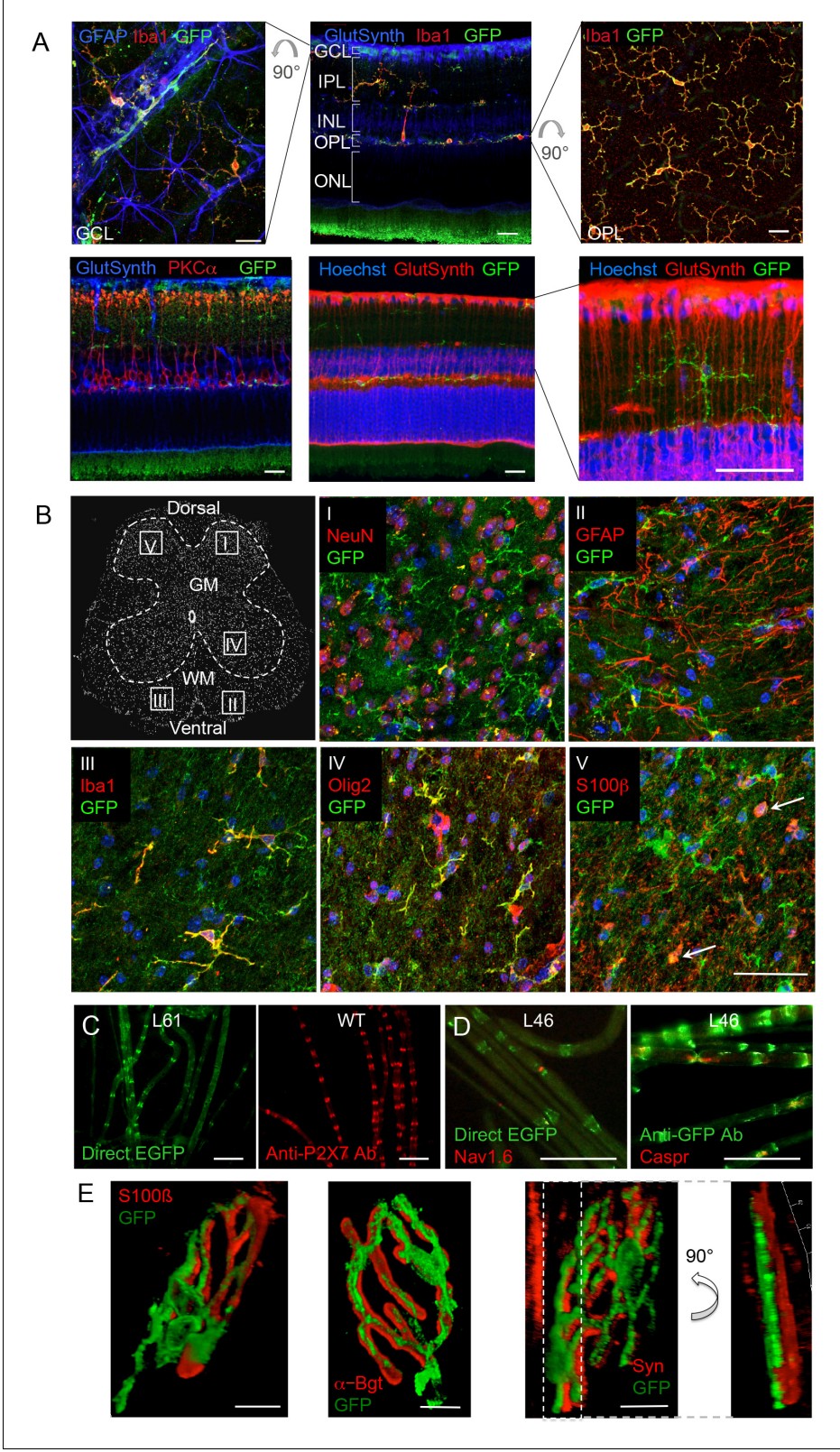

**Figure 4.** P2X7-EGFP expression in retina, sciatic nerves, spinal cord, and at the neuromuscular synapse. (**A**) EGFP exclusively co-localizes with microglia and endothelial cells in the adult mouse retina. Upper panel: *Middle*, retinal slice labeled for GFP (600 101 215, Rockland), Iba1 (marker for microglia/macrophages) and glutamine synthetase (marker for Müller glia). *Left and right*, retinal flat mounts scanned at the plane of the ganlion cell layer (GCL) and

*Figure 4 continued on next page*

*Figure 4 continued*

outer plexiform layer (OPL), respectively, to delineate microglia residing in these retinal layers. Astrocytes in the GCL were labeled with GFAP (G6171, Sigma-Aldrich). IPL, inner plexiform layer; INL, inner nuclear layer; ONL, outer nuclear layer. Lower panel: Co-staining of EGFP with neuronal marker PKCα (*left*) and glutamine synthetase (two right panels) at higher contrast and resolution to show absence of neuronal P2X7-EGFP. Cell nuclei were counterstained with Hoechst 33342 (blue) Scale bars: 20 μm. n = 2 individual line 61 in FVB/C57b/6 hybrid mice (B) Confocal images of GFP (ab6556, Abcam; A10262 or Thermo Fisher Scientific) co-immunostaining with antibodies against the indicated marker proteins in transgenic mice line 17 spinal cord slices (GFAP (MAB360, Millipore), S100β (S2532, Sigma Aldrich)). Representative images were taken from the areas shown in the schematic overview. Arrows indicate co-staining for S100ß and GFP. Scale bar: 40 μm. Cell nuclei were counterstained with DAPI (blue). Representative images from n = 3 animals are shown. (C) Comparison of transgenic P2X7-EGFP fluorescence and endogenous P2X7 immunofluorescence (P2X7 antibody, Synaptic Systems) in teased sciatic nerve fibers of line 61 and wt mice, respectively. Representative images from at least 3 animals are shown. (D) Co-staining of P2X7-EGFP (A11122, Thermo Fischer Scientific, dilution 1:1000) in teased sciatic nerve fibers of line 46 with antibodies against axonal marker proteins demonstrates localization of the transgene at perinodal regions of Schwann cells. Scale bars: 50 μm. (E) Reconstructed 3-D images of the neuromuscular junction showing co-staining of P2X7-EGFP (ab6556, Abcam; or A10262, Thermo Fisher Scientific) with perisynaptic Schwann cells (S100β (S2532, Sigma Aldrich)) as well as postsynaptic (α-Bungarotoxin, α-Bgt) and presynaptic (synaptophysin, Syn) marker proteins. The side view in the right panel shows no overlap between GFP and synaptophysin staining. Scale bars: 10 μm and 20 μm, respectively. Representative images from n = 3 animals are shown. For antibodies not specified in the legend see Key resources table.

DOI: https://doi.org/10.7554/eLife.36217.019

The following figure supplement is available for figure 4:

**Figure supplement 1.** P2X7-EGFP localization in DRG and myenteric plexus preparations.
DOI: https://doi.org/10.7554/eLife.36217.020

(*Figure 5—figure supplement 3A–D*). To test if tissue damage, that is a condition that is associated with neuroinflammation, could induce neuronal P2X7-EGFP synthesis, we proceeded our analysis with three experimental models of acute and/or invasive CNS injury: ischemic retina, stab wound, and kainic acid-induced status epilepticus. In preliminary experiments with a small number of animals (n = 3–4), an increased microglia reaction (*Figure 5A*) and microglia number (*Figure 5B*) as well as other Iba1/P2X7-positive cells (possibly invading macrophages, *Figure 5—figure supplement 1*) were observed upon transient retinal ischemia in wt animals. Interestingly, this effect appeared to be enhanced in P2X7-EGFP transgenic mice (*Figure 5A,B*). Importantly, however, P2X7 was not upregulated in other cell types than microglia, at least 3 days post injury (*Figure 5—figure supplement 1*). In this context, it should be emphasized, that a similar trend was observed in mice subjected to the stab wound injury of the somatosensory grey matter (GM). Compared to the situation in wt mice (n = 2) at 5 days after injury, post-traumatic GM of P2X7-EGFP transgenic mice (n = 3) showed a trend toward increased reactivity of microglial cells at the injury site and increased lesion area (*Figure 5C*). These data support the functionality and correct transcriptional regulation of the construct and suggest a deleterious effect of P2X7 overactivation on neurons (*Sperlágh and Illes, 2014*). The low number of animals, however, requires confirmation in future studies. Nevertheless, no induction of neuronal or astroglial P2X7-EGFP synthesis was found in the affected lesion area (*Figure 5D*), although we cannot exclude a potential obfuscation of the EGFP signal due to autofluorescence in the direct vicinity to the injury. Finally, no induction of P2X7 protein expression was observed in neurons of the dentate gyrus, CA1, and CA3 regions 24 hr after induction of status epilepticus by a unilateral intra-amygdala kainic acid injection, although a change of microglia morphology clearly indicated their activation (*Figure 5—figure supplement 2*). In conclusion, we suggest that P2X7-dependent neurodegeneration that has been observed in various studies is caused by an indirect mechanism, most likely involving P2X7 activation in microglia or oligodendrocytes.

## Discussion

P2X7 research has suffered from a lack of specific and sensitive antibodies. While some antibodies detect P2X7 in western blots and upon overexpression in cultured cells, they fail to localize it reliably in complex preparations such as brain sections. In addition, an intricate pharmacology and

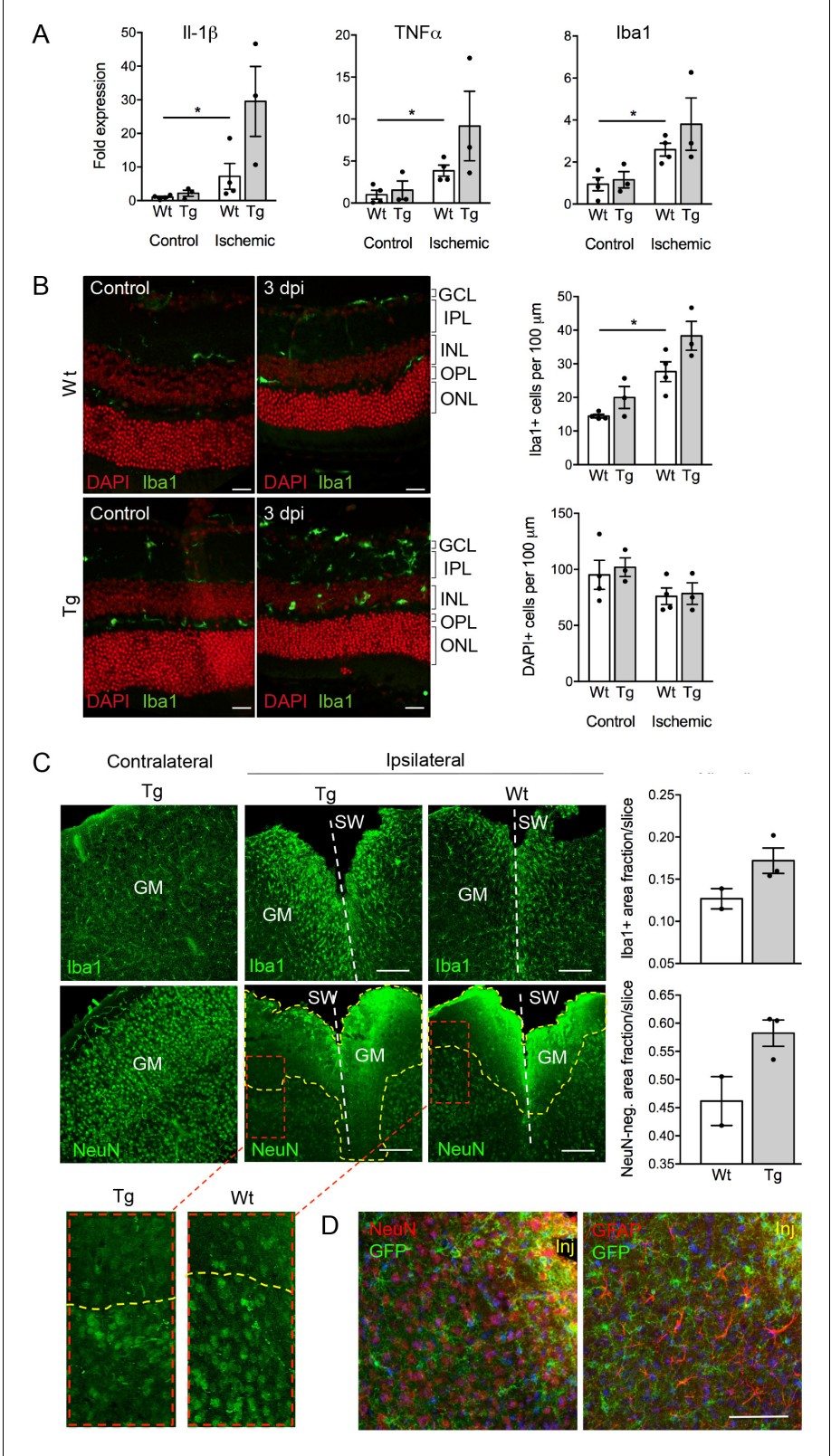

**Figure 5.** Consequences of P2X7 overexpression under physiological and pathophysiological conditions. (**A**) Quantitative real-time PCR was performed in duplicates on samples from microglia isolated by immunomagnetic separation from control and postischemic (3 days post injury, 3 dpi) retinae. Bars represent mean ±SEM and include data from 3-4 animals/each genotype/condition. Significance between expression level in the untreated

*Figure 5 continued on next page*

*Figure 5 continued*

control eye of the respective genotype was analyzed using unpaired two-tailed Mann-Whitney-U-test and indicated as: *p<0.05. (**B**) Retinal slices labeled for the microglia/macrophage marker Iba1. Cell nuclei were counterstained with DAPI (red). Retinae were isolated from mice of which one eye had been subjected to transient ischemia. The untreated contralateral eye served as internal control. Dpi, days post-injury. Scale bars: 20 µm. Cell numbers of the inner retinal layers and microglia specifically were quantified in 2–5 central retinal slices per animal on basis of DAPI and Iba1 staining, respectively. Bars represent mean ±SEM and include data from 3 to 4 animals/ genotype/condition. Note that data from transgenic mice were not significantly different in A and B. (**C**) Representative confocal images of coronal sections from posttraumatic GM at 5 dpi. Slices of the somatosensory grey matter (GM) from wt and transgenic animals stained for NeuN- (neurons) and Iba1- (microglia) positive cells are shown. White dotted lines indicate stab wounds; yellow dotted lines indicate NeuN-negative lesion areas. Insets show chosen borders between NeuN-positive and negative areas. Bar diagrams depict fractions of Iba1-positive and NeuN-negative areas in relation to DAPI-positive areas. Means of N = 2–3 animals (n = 6–7 sections per animal)±SEM are shown. Scale bar: 200 µm. One wt tissue broke and could not be analyzed. (**D**) Double immunostaining with GFP (ab6556, Abcam) and NeuN or GFAP (for astroglia (MAB360, Millipore)) shows no upregulation of P2X7-EGFP in these cell types within the penumbra of P2X7-EGFP and wt mice at 5 dpi. Note that immunofluorescence in the area immediately adjacent to the lesion core (~0–75 µm) is non-specific due to autofluorescence of cells within damaged tissue (inj), and this might obfuscate a potential P2X7-EGFP signal. Cell nuclei were counterstained with DAPI (blue). Scale bar: 100 µm. For antibodies not specified in the legend see Key resources table.

DOI: https://doi.org/10.7554/eLife.36217.021

The following figure supplements are available for figure 5:

**Figure supplement 1.** Upregulation of P2X7 expression in postischemic retinae 3 days post injury (dpi) of wt and P2X7-EGFP transgenic animals.
DOI: https://doi.org/10.7554/eLife.36217.022

**Figure supplement 2.** Cell type-specific P2X7-EGFP expression in the three hippocampal subfields (DG, CA1, and CA3) upon induction of status epilepticus.
DOI: https://doi.org/10.7554/eLife.36217.023

**Figure supplement 3.** Effects of P2X7 overexpression under physiological conditions.
DOI: https://doi.org/10.7554/eLife.36217.024

compromised *P2rx7*$^{-/-}$ models (*Masin et al., 2012*; *Nicke et al., 2009*) complicate the analysis of its localization and function *in vivo*. Here, we present a novel P2X7-EGFP BAC transgenic mouse model that overexpresses functional fluorescence-tagged P2X7 and is able to specifically report P2X7 expression at the protein level. Moreover, this model permits studies of the functional consequences of P2X7 overexpression. Detailed analyses of these mice under physiological conditions show that in the CNS, P2X7 is predominantly located in microglia and oligodendrocytes and to a minor extent in a fraction of S100β-positive cells in the cerebrum as well as Bergmann glia in the cerebellum. Given this distribution in physiology and the fact that no upregulation of P2X7 protein in neurons was observed after neural tissue damage or following status epilepticus, it is conceivable that the reported P2X7-dependent neuronal damage is the consequence of the pronounced manifestation of microglia activation rather than direct activation of neuronal P2X7 receptors.

## The BAC transgenic approach

Numerous examples demonstrate that BAC transgenics are valuable tools to investigate endogenous protein expression patterns (*Gerfen et al., 2013*; *Yang and Gong, 2005*). In comparison to knock-in approaches, they provide the advantages of a stronger signal due to the moderate overexpression which might boost physiological functions and thus make them accessible for an in depth analysis. Together with classical and/or conditional knock-out strategies, this provides a powerful combination for the *in vivo* analysis of protein functions. In contrast to small transgenes, in which the expression patterns are often affected by the integration site, BAC transgenes show in most cases an expression pattern that reflects the endogenous promoter within the BAC. However, position effects such as gene deletion or aberrant expression due to integration of (truncated) BAC transgenes in other genes or regulatory elements cannot be excluded. Thus, five transgenic lines were analyzed in detail in this study, and all showed identical expression patterns. A correct expression pattern was further corroborated by 1) a comparative analysis of P2X7-EGFP transgene expression

and endogenous P2X7 expression in wt mice using a novel P2X7-specific nanobody-rbIgG fusion construct and 2) cell type-specific P2X7 deletion in oligodendrocytes and microglia using conditional P2X7 knock-out mouse models. Together, these findings all argue against an ectopic P2X7-EGFP expression pattern and indicate a predominant expression of P2X7 in non-neuronal cells within the brain parenchyma.

## P2X7 protein localization in neurons?

The absence of a neuronal localization of P2X7-EGFP contrasts with findings in a BAC transgenic reporter mouse line (Tg(P2rx7 EGFP)FY174Gsat; GenSat) in which soluble EGFP is expressed under the control of a BAC transgenic P2X7 promoter and a recently described humanized P2X7 knock-in mouse model (*Metzger et al., 2017*). Both models show neuronal P2X7 expression but differ remarkably in the observed expression patterns (*Engel et al., 2012*; *Jimenez-Pacheco et al., 2013*; *Metzger et al., 2017*). Neuronal expression of the soluble EGFP reporter in the GenSat mouse is seen in multiple brain regions whereas P2X7 transcripts in the humanized P2X7 mouse model seem to predominate in CA3 neurons (*Metzger et al., 2017*) (*Table 2*). In contrast to our findings at the protein level, the knock-in model showed only a minor reduction in P2X7 expression in microglia-specific knock-out animals. A possible explanation for these discrepancies could be that alterations in gene structure introduced into the GenSat P2X7 BAC EGFP mice influence post-transcriptional and translational regulatory mechanisms. For example, intron 1, the importance of which is evidenced by the P2X7 k splice variant (*Nicke et al., 2009*), is not fully preserved in the EGFP reporter mouse and soluble EGFP is translated from a truncated mRNA which might lack regulatory elements. In case of the humanized P2X7 model, only RNA transcripts were analyzed which might not correlate with protein synthesis. Moreover, different sensitivities of the detection methods (*in situ* hybridization versus immunohistochemistry) could also account for some of the observed differences. As even minor gene modifications, such as the flanking of the exons with the comparatively small loxP sequences, are able to influence gene expression (*Requardt et al., 2009*; *Zhang et al., 2013*) (*Figure 3H*), we kept the *P2rx7* gene structure almost untouched and retained as much of the UTRs as possible to avoid such unpredictable influences. Although we cannot completely rule out an effect of the introduced gene modification within the BAC or a loss of possible modulatory elements that lie outside of the chosen BAC, comparison of the BAC transgenic P2X7-EGFP with endogenous P2X7 expression as analyzed with the novel nanobody-based heavy chain antibody, provide strong evidence that the expression pattern observed in our mouse line matches that of the endogenous protein. This, however, does not preclude species differences. For example, in human but not in murine Müller cells, functional and immunohistological evidence for P2X7 expression has been shown (*Franke et al., 2005*; *Innocenti et al., 2004*; *Pannicke et al., 2000*).

Finally, we cannot exclude neuronal expression that is below the detection limit. As the P2X7 receptor is known to be a highly regulated protein and has been shown to be deleterious to cultured neuronal cells (*Ohishi et al., 2016*), its expression and localization should be tightly regulated in post-mitotic cells like neurons. If present in neurons, its presence would likely be limited to subcellular regions were synapse formation and selection takes place and/or to areas where damaged cells need to be removed by apoptosis. Possible extrasynaptic or growth cones localizations (*Díaz-Hernandez et al., 2008*) would be difficult or impossible to resolve *in situ* using the described antibodies and conventional microscopy. However, even upon induction of tissue damage, virtually no P2X7 expression in cerebral cell types other than microglia or oligodendrocytes was observed. Based on our data, we therefore suggest that P2X7-induced neurodegeneration is due to an indirect effect (i.e. extended glial reaction within the acute post-traumatic period), which requires further investigation.

## Can P2X7 splice variants account for neuronal P2X7 expression?

Five murine P2X7 splice variants (*Masin et al., 2012*; *Nicke et al., 2009*) have been identified. Two of these (variants a and k) contain the C-terminal sequence that was fused to EGFP and should be detectable in our mouse model. The other three variants (b, c, and d) are C-terminally truncated or altered and could therefore escape the detection by both our EGFP-tag and the most commonly used antibodies against the P2X7 C-terminus. However, the nanobody used in our study was selected on intact P2X7 expressing cells and therefore binds at the extracellular P2X7 domain which

is present in splice variants b and c. Together with the fact that in the original study (**Masin et al., 2012**), protein of the deleted variants was not detected in the brain, this strongly argues against the presence of the mouse P2X7 variants b and c in neurons. Splice variant d contains only one TM domain and cannot be expected to form functional receptors. We therefore consider it highly unlikely that one of the known mouse variants of P2X7 accounts for neuronal P2X7 function or detection. A 65 kD protein band is frequently detected by us and others with antibodies against the P2X7 C-terminus but, unlike the P2X7 protein (variant a, about 75 kD), does not show a size shift upon glycosidase treatment and therefore, most likely does not represent a P2X7 variant.

In summary, we generated and thoroughly characterized a novel transgenic P2X7-EGFP mouse model that should help to overcome previous limitations in P2X7 research. Using functional assays, we could show that transgenic expressed P2X7-EGFP rescues the phenotype of P2X7 deficiency thus confirming the functionality of the transgene. Our initial characterization, however, indicates that P2X7 overexpression does not per se induce any overt pathologies, but rather represents a natural response observed after tissue damage. This is in line with observations in Schwann cells, where overexpression of P2X7 alone does not alter the basal $Ca^{2+}$ concentration: overexpression of the peripheral myelin protein 22 (as it occurs in Schwann cells from patients suffering from Charcot Marie Tooth type 1A), causes instead a secondary overexpression of P2X7 and consequent Schwann cells functional derangement (**Nobbio et al., 2009**), suggesting that in addition to P2X7 overexpression, other factors are required to induce P2X7-associated pathologies. Most importantly for our study, neuronal P2X7 protein expression was not induced under pathological conditions. An unresolved question is, whether the high ATP concentrations required to activate the receptor occur physiologically or if the receptor is silent under physiological conditions and mainly expressed and/or activated under pathophysiological conditions. The presented mouse model provides a new tool for addressing this question.

# Materials and methods

## Key resources table

| Type | Designation | Source or reference | Identifiers | Additional information |
|---|---|---|---|---|
| BAC clone | BAC clone, RP24-114E20 | Children's Hospital Oakland Research Institute, Oakland, CA | | Strep-tagII-Gly-7xHis-Gly-EGFP-sequence was inserted into the $P2r \times 7$ BAC clone RP24-114E20 |
| Strain (*Mus musculus*) | P2X7 EGFP (FVB/N-Tg(RP24-114E20P2X7 StrepHis-EGFP)Ani) | this paper | | Lines: 46, 59 (also in BL/6N), 61 |
| Strain (*Mus musculus*) | P2X7[451P]-EGFP (FVB/N-Tg(RP24-114E20P2X7451P-StrepHis-EGFP)Ani) | this paper | | Lines: 15, 17 (also in BL/6N) Transgenes were backcrossed into C57BL/6 for at least eight generations |
| Strain (*Mus musculus*) | B6-P2rx7[tm1a(EUCOMM)Wtsi] | European Mutant Mouse Archive | MGI:4432150 | |
| Strain (*Mus musculus*) | Gt(ROSA)26Sor[tm1(FLP1)Dym] | Farley FW, et al., Genesis. 2000 | MGI:2429412 | FLPe deleter |
| Strain (*Mus musculus*) | Tg(EIIa-cre)C5379Lmgd | Lakso M, et al., Proc Natl Acad Sci U S A. 1996 | MGI:2137691 | EIIa-Cre mouse |
| Strain (*Mus musculus*) | *B6-Cx3cr1[tm1.1(cre)Jung]* | Yona S, et al., Immunity. 2013 | MGI:5467983 | Cx3cr-1-Cre |
| Strain (*Mus musculus*) | B6- Cnp[tm1(cre)Kan] | Lappe-Siefke C, et al., Nat Genet. 2003 | MGI:3051635 | CNP-Cre |
| Antibody | P2X7 C-term (rb pAb) | Synaptic Systems | Cat# 177003, RRID: AB_887755 | WB 1:1500 IHC 1:500 |
| Antibody | P2X7 ECD (rb pAb) | Alomone | Cat# APR-008 RRID:AB_2040065 | WB 1:500 IHC 1:500 |
| Antibody | ß-Actin (ms AC-15) | Sigma-Aldrich | Cat# A3854 RRID:AB_262011 | WB 1:15.000 |

*Continued on next page*

Continued

| Type | Designation | Source or reference | Identifiers | Additional information |
|------|-------------|---------------------|-------------|------------------------|
| Antibody | Vinculin (ms hVin-1) | Sigma-Aldrich | Cat# V9131 RRID:AB_477629 | WB 1:10.000 |
| Antibody | 800CW gt anti-ms | LI-COR | Cat# 925–32210 RRID:AB_2687825 | WB 1:15.000 |
| Antibody | 680RD dk anti-rb | LI-COR | Cat# 925–68073 RRID:AB_2716687 | WB 1:15.000 |
| Antibody | 680RD gt anti-rb | LI-COR | Cat# 925–68071 RRID:AB_2721181 | WB 1:15.000 |
| Antibody | CD11b-perCP (rat M1/70) | BioLegend | Cat 101230, RRID:AB_2129374 | FACS 1:100 |
| Antibody | CD45-PE-Cy7 (rat 30-F11) | BioLegend | Cat 103114, RRID:AB_312979 | FACS 1:100 |
| Antibody | CD16/32 (Fc-Block, rat 2.4G2) | BioXcell | Cat# BE0307 RRID:AB_2736987 | FACS 1:100 |
| Antibody | P2X7 C-term. (rb pAb) | Alomone | Cat# APR-004 RRID:AB_2040068 | IHC 1:500 |
| Antibody | P2X7 ECD, 7E2-rbIgG | Nolte lab | Nanobody rbIgG fusion construct | (0.1 ug/ml) |
| Antibody | GFP (rb pAb) | Abcam | Cat# ab6556 RRID:AB_305564 | IHC 1:2000 |
| Antibody | GFP (chk pAb) | Thermo Fisher | Cat# A10262, RRID:AB_2534023 | IHC 1:400 |
| Antibody | GFP (rb pAb) | Thermo Fisher | Cat# A6455, RRID:AB_221570 | IHC 1:250 |
| Antibody | GFP (rb pAb) | Thermo Fisher | Cat# A11122, RRID:AB_221569 | IHC 1:400 |
| Antibody | GFP (gt pAb) | Rockland | Cat# 600-101-215 RRID:AB_218182 | IHC 1:200 |
| Antibody | MAP2 (ms 198A5) | Synaptic Systems | Cat# 188011, RRID: AB_2147096 | IHC 1:500 |
| Antibody | NeuN (ms A60) | Millipore | Cat# MAB377 RRID:AB_2298772 | IHC 1:500 |
| Antibody | GFAP (ms GA5) | Millipore/Sigma-Aldrich | Cat# MAB360/G6171 RRID:AB_11212597/ AB_1840893 | IHC 1:200/500 |
| Antibody | GFAP (rb pAb) | Dako | Cat# Z0334, RRID:AB_10013382 | IHC 1:1000 |
| Antibody | S100ß (rb pAb) | Synaptic Systems | Cat# 287003, RRID: AB_2620024 | IHC 1:500 |
| Antibody | S100ß (rb pAb) | Abcam | ab7853 (not longer available) | IHC 1:1000 |
| Antibody | S100ß (ms SHB1) | Sigma-Aldrich | Cat# S2532, RRID:AB_477499 | IHC 1:400 |
| Antibody | Iba1 (rb pAb) | WAKO | Cat# 019–19741 RRID:AB_839504 | IHC 1:100 |
| Antibody | Olig 2 (ms 211F1.1) | Millipore | Cat# MABN50 RRID:AB_10807410 | IHC 1:200 |
| Antibody | NG2 (rb pAb) | Millipore | Cat# AB5320 RRID:AB_91789 | IHC 1:500 |
| Antibody | VGAT (ms CL2793) | Molecular Probes | Cat# MA5-24643 RRID:AB_2637258 | IHC 1:200 |
| Antibody | vGlut1 (ms 317G6) | Synaptic Systems | Cat# 135511, RRID: AB_887879 | IHC 1:100 |

*Continued on next page*

*Continued*

| Type | Designation | Source or reference | Identifiers | Additional information |
|---|---|---|---|---|
| Antibody | vGlut2 (rb pAb) | Synaptic Systems | Cat# 135 403, RRID:AB_887883 | IHC 1:100 |
| Antibody | PSD95 (ms 108E10) | Synaptic Systems | Cat# 124011, RRID:AB_10804286 | IHC 1:100/500 |
| Antibody | Calretinin (ms 37C9) | Synaptic Systems | Cat# 214111, RRID: AB_2619904 | IHC 1:1000 |
| Antibody | Calbindin D28k (ms 351C10) | Synaptic Systems | Cat# 214011, RRID:AB_2068201 | IHC 1:200 |
| Antibody | Calbindin D28k (gp pAb) | Synaptic Systems | Cat# 214 005, RRID:AB_2619902 | IHC 1:100 (only used for data confirmation, not in manuscript) |
| Antibody | Synaptophysin (ms pAb) | Synaptic Systems | Cat# 101011, RRID:AB_887824 | IHC 1:500 |
| Antibody | Nav 1.6 (rb pAb) | Alomone | Cat# ASC009 RRID:AB_2040202 | IHC 1:100/500 |
| Antibody | Caspr (ms K65/35) | Neuromab | Cat# 75–001 RRID:AB_2083496 | IHC 1:1000 |
| Antibody | Cre (ms 2D8) | Millipore | Cat# MAB3120 RRID:AB_2085748 | IHC 1:200 |
| Antibody | ß3-Turbulin (gp pAb) | Synaptic Systems | Cat# 302304 RRID:AB_10805138 | IHC 1:200 |
| Antibody | GlutSynth (ms GS-6) | Millipore | Cat# MAB302 RRID:AB_2110656 | IHC 1:500 |
| Antibody | PKCα (rb Y124) | Abcam | Cat# ab32376, RRID:AB_777294 | IHC 1:200 |
| Antibody | TH (rb pAb) | Millipore | Cat# AB152 RRID:AB_390204 | IHC 1:200 |
| Antibody | Nestin (ms rat-401) | Millipore | Cat# MAB353 RRID:AB_94911 | IHC 1:100 |
| Antibody | Sox9 (rb pAb) | Novus bio | Cat# NBP1-85551-25 RRID:AB_11002706 | IHC 1:100 |
| Antibody | MPZ (rb pAb) | Abcam | Cat# ab31851, RRID:AB_2144668 | IHC 1:200 |
| Antibody | Hu C/D (ms 16A11) | Thermo Fisher | Cat# A-21271, RRID:AB_221448 | IHC 1:200 |
| Antibody | A594 gt anti-ms | Thermo Fisher | Cat# A11032 RRID:AB_2534091 | IHC 1:400 |
| Antibody | A594 gt anti-rat | Thermo Fisher | Cat# A11007, RRID:AB_10561522 | IHC 1:400 |
| Antibody | A546 gt anti-ms | Thermo Fisher | Cat# A-11003, RRID:AB_2534071 | IHC 1:400 |
| Antibody | A488 gt anti-rb | Thermo Fisher | Cat# A11008, RRID:AB_143165 | IHC 1:400 |
| Antibody | A488 gt anti-chk | Thermo Fisher | Cat# A11039, RRID:AB_2534096 | IHC 1:400 |
| Antibody | A633 gt anti-rb | Thermo Fisher | Cat# A21070, RRID:AB_2535731 | IHC 1:400 |
| Antibody | A633 gt anti-gp | Thermo Fisher | Cat# A21105, RRID:AB_1500611 | IHC 1:400 |
| Antibody | Cy3 gt anti-rb | Jachkson Res. | Cat# 111-165-003, RRID:AB_2338000 | IHC 1:300 |
| Antibody | Cy3 gt anti-ms | Jachkson Res. | Cat# 115-165-146, RRID:AB_2338690 | IHC 1:300 |

## Analysis of the mP2X7-EGFP constructs upon expression in HEK cells

HEK293 cells were cultured and transiently transfected with 1.5-2 mg DNA/well of a 6-well-plate (Lipofectamin, Thermo Fisher Scientific). After 48 h, cells were washed and collected in PBSs (2 wells/experiment), pelleted in a desktop centrifuge (2'at 13,000 rpm) and extracted as described (Nicke et al., 2008) for 15 minutes on ice in 100 ml 0.1% sodium phosphate buffer (pH 8.0) containing 1% digitonin (Fluka, Buchs, Switzerland) and 0.4 mM Pefabloc SC (Fluka). 10ml of extract were separated by SDS-PAGE with with or without endoglycosidase treatment (30 min at 37 °C in the presence of reducing loading buffer (1x) and 5 IUB miliunits EndoH or 10 IUB miliunits PNGaseF (New England Biolabs)). For ethidium uptake measurements, cells were seeded after 27 h in 96-well plates ($5 \times 10^4$ cells/well) and incubated in the presence of 20 mM ethidium bromide in PBS for 15 min. Dye influx was evaluated with a fluorescence plate reader (Fluostar Galaxy, BMG) upon addition of the indicated ATP concentrations, as described (Bruzzone et al., 2010). Patch-clamp recordings were performed as described (Nicke et al., 2009) in normal (147 mM NaCl, 2 mM KCl, 2 mM $CaCl_2$, 1 mM $MgCl_2$, 10 mM HEPES, and 13 mM glucose) or low divalent cation (0 $MgCl_2$, 0.1 mM $CaCl_2$) containing extracellular solution.

## Generation of mP2X7-EGFP BAC transgenic mice (FVB/N-Tg(RP24-114E20-P2X7/StrepHisEGFP))

The Strep-tagII-Gly-7xHis-Gly-EGFP-sequence was inserted into the *P2rx7* BAC clone RP24-114E20 (Children's Hospital Oakland Research Institute, Oakland, CA), immediately upstream of the *P2rx7* stop codon by homologous recombination (Warming et al., 2005) using locus-specific homology arms of 50–60 bp length (Expand High Fidelity PCR, Roche Applied Science). The 451P variant was generated from the obtained BAC by the same strategy. BAC DNAs were verified after each recombination step by PCR, Southern blot and DNA fingerprinting. Upon sequencing of the coding regions (Eurofins Genomics, Germany), final BAC constructs were linearized with SacII (thereby destroying the unwanted Ift81 gene), purified (Sepharose CL-4B chromatography, Sigma-Aldrich), analyzed by pulsed field gel electrophoresis, and microinjected into pronuclei of FVB/N mouse zygotes (451L background) (for primers and probes see *Supplementary files 1* and *2*).

## Southern blot analysis

Genomic DNA was isolated from tail biopsies, digested with BglII, separated on an 0.8% agarose gel, and blotted onto Nylon membrane (Hybond N+, GE Healthcare) by capillary transfer. After immobilization by UV irradiation (1500 $\mu J/cm^2$), DNA was hybridized to a $^{32}P$ labeled probe (Random primed labeling Kit, Roche) corresponding to a 645 bp fragment 2.6 kb downstream of the *P2rx7* stop codon. Autoradiographic analysis (Phosphoimager plates, Molecular Dynamics) specifically detected the expected hybridization signals at 5277 bp (BAC transgene) and 4561 bp (endogenous *P2rx7*). The intensity ratios were used to determine the number of inserted BAC copies (for probes see *Supplementary file 1*).

## Biochemistry

Protein extracts were prepared as described (Nicke, 2008) using a Precellys homogenizer (Peqlab) with CK28 beads (15 s, 5.000 rpm) and NP40 (Sigma) as detergent. Protein concentrations were determined by BCA assay (Pierce). 30–75 µg of total protein per lane were loaded on 8% SDS-PAGE gels. Protein was either directly visualized by EGFP fluorescence scanning (Typhoon, GE Healthcare) or blotted onto Immobilon-FL PVDF membranes (Merck Millipore) and detected with an Odyssey infrared imaging system (LI-COR Biosciences) using the indicated antibodies (S1 Material and methods). Endoglycosidase (New England Biolabs) treatment was performed for 30 min at 37°C in 20 µl sample aliquots with loading buffer (IUB miliunits: EndoH 10, PNGaseF 20).

## FACS analysis of microglia dye uptake

Mice (8–12 weeks, male and female) were sacrificed and single-cell suspensions prepared from brains by 30 min collagenase digestion at 37°C in a shaking water bath. Cell suspensions were filtered through a 70 µm cell strainer and centrifuged for 5 min at 300x g. Microglia were separated from debris by resuspending the pellet in 5 ml of a 33% Percoll solution (GE Healthcare) and centrifugation (20 min, 300x g). The pellet was resuspended in 1 ml ammonium-chloride-potassium

erythrocyte lysis buffer and incubated for 1 min on ice to remove erythrocytes. Cells were subsequently washed with 10 ml FACS buffer (PBS + 0.2% BSA/1 mM EDTA) and resuspended in 100 µl FACS buffer. Microglia were stained (30 min on ice) with anti-CD11b-perCP (Biolegend) and anti-CD45-PE-Cy7 (Biolegend) in the presence of Fc-blocking anti-CD16/CD32 (BioXcell) and normal rat serum. After washing 2x with FACS buffer cells were resuspended in 200 µl RPMI medium (Gibco). DAPI was added to a final concentration of 1.5 µM and cells were incubated in the presence or absence of 1 mM ATP at 37°C for 15 min. The DAPI uptake into CD11b$^+$CD45$^{low}$ microglia was subsequently measured using a BectonDickinson Celesta flow cytometer.

For monitoring of time-dependent DAPI uptake by real time flow cytometry, isolated microglia were differentiated by transgenic P2X7-EGFP expression and exogenous eFluor$^{670}$-labeling (WT or P2X7 knockout) and pooled in one FACS tube in 500 µl RPMI medium (Gibco) in the presence of DAPI (1.5 µM) in order to have identical stimulation conditions. The baseline DAPI signal was measured for 2 min at 37°C, then 1 mM ATP was added and measuring continued for 4 to 5 min. DAPI uptake over time was compared among the differentially labeled microglia.

## Generation of P2X7-specific nanobody-based heavy chain antibody 7E2-rbIgG

The coding region for the P2X7-specific nanobody 7E2 was cloned into the pCSE2.5 vector (provided by T. Schirrmann, Technical University Braunschweig, Germany) (*Schirrmann and Büssow, 2010*) upstream of coding regions for the hinge, CH2 and CH3 domains of rabbit IgG (*Danquah et al., 2016*). Six days after transfection of this construct into HEK-6E cells (*Zhang et al., 2009*), 7E2-rbIgG was purified from the cell supernatant by affinity chromatography on a protein-G sepharose column. Buffer was exchanged by gel filtration on a PD-10 column. A panel of nanobody-rbIgG heavy chain antibodies was originally screened for binding to P2X7 transfected HEK cells before and after fixation with 4% PFA. 7E2-rbIgG was chosen because it retained the strongest staining after fixation in both immunofluorescence staining and a FACS-based dissociation assay analogous to that described in (*Fumey et al., 2017*). It only weakly antagonizes gating of P2X7 by ATP and by ADP-ribosylation but its potency was not further determined (*Danquah et al., 2016*).

## Diaminobenzidine (DAB) immunohistochemistry

Mice were sacrificed by $CO_2$/cervical dislocation or anesthetized (Ketamin/Xylazin) and transcardially perfused with 4% PFA. Brains were fixed in 4% PFA for 72 hr or 24 hr, respectively, cryoprotected in 30% sucrose, and embedded in 5% LM Agarose (Roth, Germany). 30 µm sagittal brain sections were prepared (VT1200s Leica Microsystems) and either blocked with 4% skim milk and 10% FCS in PBS (1–1.5 hr, RT) or, after peroxidase block (3% $H_2O_2$ in 0.01 M PBS, 30 min RT), with 10% Normal Goat Serum and 0.1% Triton X-100 in PBS (1 hr, RT), prior to primary antibody incubation overnight at 4°C. Incubation with biotinylated secondary antibodies was at 37°C for 1 hr, or at RT for 1.5 hr. Staining was visualized using the ABC method with the Vectastain abc kit and the DAB substrate kit for peroxidase (Vectorlabs, USA) or SIGMA_FAST_ DAB Tablets (Sigma-Aldrich, Germany). Counterstaining was carried out with hematoxylin (Sigma-Aldrich), followed by dehydration and embedding. Images were taken with an Axio Observer 7 (Zeiss).

## Immunofluorescence staining

Immunostaining was performed as described (*Zhang et al., 2013*). In brief, mice (P60-P90) were transcardially perfused with PBS and then 4% PFA in PBS. Brains or spinal cords were post-fixed over night in 4% PFA/PBS. P7 pups were decapitated and brains post-fixed in 4% PFA. After cryoprotection (30% sucrose in PBS (pH 7.4), 40 µm cryostat sections (Microm HM560, Walldorf, Germany) were washed (3 × 10 min, PBS), blocked (5% Normal Goat Serum (Dako Germany), 0.3% Triton X-100 (Sigma, Munich, Germany) in PBS, 2 hr at RT) and incubated with primary antibodies (16–24 hr, 4°C) with gentle shaking. After washing as above, sections were incubated for 2 hr with fluorescence conjugated secondary antibodies. Slices were mounted on object slides with Perma-Fluor Mounting Medium (Thermo Scientific).

*Rectus femoris* muscle was incubated in 30% sucrose (48 hr), embedded in OTC (Tissue TEK), and frozen in liquid nitrogen. 20 µm cryostat sections were collected on object slides, fixed with 4% PFA (10 min), blocked and permeabilized (30 min, 1% BSA, 0.2% Triton X-100 in PBS). Incubation with

Alexa 594 Fluor-conjugated α-bungarotoxin (1:1000, Thermo Fisher Scientific) and primary antibodies was overnight at 4°C, and with secondary antibodies for 1 hr at RT.

Thoracic and lumbar DRGs were embedded in Tissue-Tek, sectioned in 10 µm slices, mounted on slides and frozen at −80°. Before staining as described above, sections were post-fixed for 10 min in 4% PFA in PBS, incubated for 30 min in 0.1 M Glycine in PBS, and blocked for 1 hr (5% normal goat serum, 0.1% Triton X-100 in PBS). In case of glutamine synthetase and MPZ staining, tissue was treated with 10 mM sodium citrate (pH 6.0,>95° C) for 1 min just before blocking. Images were obtained by confocal laser scanning microscopy (Leica SP5 or Zeiss LSM 880).

### Quantification and statistical analysis

EGFP-positive cells were quantified in every fifth slice in a series of 25–30 sections throughout the whole rostrocaudal extension of the hippocampus. DAPI-positive cells, EGFP-positive cells, and marker protein-positive cells in the hippocampus CA1 region were counted in z-stacks. To define the counting box ($250 \times 250 \times 25$ µm$^3$), confocal laser micrographs of the CA1 region were obtained (63 x/0,75 NA objective) at 1 µm intervals to a final depth of 25 µm. Cell nuclei located completely inside the counting frame and at the upper and right borders were counted. Data analysis was performed using Excel and Graphpad Prism 7 software. Data are given as mean ±SEM from N = 3 mice per group.

### Dissociation of adult cerebellar tissue

Mice were killed by cervical dislocation and brains were rapidly removed, washed with ice cold PBS, and kept on ice. Cerebelli were homogenized using the GentleMacs neuronal tissue dissociation kit (T) (Miltenyi Biotech) according to the manufactures instruction. Dissociated cells were centrifuged (1000 g/ 15 min/4°C), and washed twice with PBS (followed by centrifugation as above) to remove residual trypsin. Supernatant was carefully removed and cells fixed (4% PFA in PBS, 10 min, 4°C) under gentle agitation, washed three times with PBS as described above, permeabilized with blocking solution (2% BSA, 2% normal goat serum, 0.2% Triton X-100 in PBS), and incubated with the indicated antibodies diluted in blocking solution overnight at 4°C. After washing, cells were incubated with secondary antibodies for 1 hr, washed, incubated with DAPI (200 nM in PBS, 10 min), washed, and embedded in Aquamount (Polyscience). Imaging was performed using a Zeiss Confocal microscope (LSM 800) and the ZEN imaging software. In co-processed wild-type animals, no GFP (immuno)-fluorescence was detected.

### Teased fiber preparation and staining

Sciatic nerves of adult mice were dissected and transferred into cold PBS. Under a stereomicroscope, the epineurium was carefully removed, nerves separated longitudinally into individual or small bundles of fascicles, transferred to a droplet of cold PBS on a superfrost slide, and gently teased apart. Samples were air-dried and stored at −20°C if not processed immediately for immunocytochemistry. Preparations were post-fixed (5 min) in 4% PFA, permeabilized with ice-cold methanol (5 min), washed with PBS (3 × 5 min), and blocked (10% horse serum, 0.1% Tween 20 in PBS, 2 hr at RT). Slides were incubated overnight at 4°C with the primary antibodies and after washing with PBS (3 × 5 min), secondary antibodies were applied (2 hr at RT). After final washing, fibers were mounted with Vectashield Mounting Medium containing DAPI (Vector Laboratories).

### Histological and immunohistochemical staining of retinae

Retinae were immersion-fixed (4% PFA for 2 hr), washed with PBS, sucrose cryoprotected and cut in 20 µm thick sections. Retinal sections were permeabilized (0.3% Triton X-100 plus 1.0% DMSO in PBS) and blocked (5% normal goat serum with 0.3% Triton X-100 and 1.0% DMSO in PBS, 2 hr at RT). Primary antibodies were incubated overnight at 4°C. Sections were washed (1% BSA in PBS) and incubated with secondary antibodies (2 hr at RT). Cell nuclei were labeled with DAPI (1:1000; Life Technologies). Control experiments without primary antibodies showed no unspecific labeling. Images were acquired using confocal microscopy (Visiscope, Visitron Systems).

For quantification of cell numbers or microglia only central retinal slices were used. Cells were quantified in a defined area of 100 µm in width (DAPI staining) or the whole scan field (~460 µm in

width; microglia) approximately 200–300 µm in distance from the optic nerve head. For each animal, 2–3 central slices were analyzed.

## Myenteric plexus preparation and staining

For whole mount myenteric plexus preparation, mice were sacrificed by cervical dislocation and 1 cm segments were taken 1 cm distal from the proximal colon and transferred to ice-cold Krebs solution (containing in mM: 117 NaCl, 4.7 KCl, 1.2 $MgCl_2$, 1.2 $NaH_2PO_4$, 25 $NaHCO_3$, 2.5 $CaCl_2$, 11 glucose and aerated with carbogen to pH 7.4) in Sylgard (Dow Corning)-filled dissecting dishes. After flushing with Krebs buffer, segments were opened along the mesenteric border, pinned out, and fixed for 4 hr at 4°C (4% PFA and 0.2% picric acid in 0.1 M phosphate buffer (pH 7.4). Tissue was rinsed (3 × 10 min) with phosphate buffer and dissected in PBS (pH 7.4). After carefully removing mucosa, submucosa and circular musculature using forceps and a binocular, myenteric plexus preparations were blocked (0.5%, Triton X-100, 0.1% $NaN_3$, 4% goat serum (Sigma-Aldrich) in PBS, 1 hr, RT), incubated with primary antibodies in the above solution (12 hr at RT), rinsed 3 × 10 min with PBS, and incubated with secondary antibodies (2 hr at RT). After washing (3x in PBS) preparations were mounted in PermaFluor (Thermo Fisher Scientific) on slides and images were acquired with a Zeiss 880 Airyscan confocal microscope and processed with ImageJ.

## Magnetic-activated cell sorting (MACS) of retinal cells

For enrichment of distinct retinal cell types a previously described protocol was used with minor modifications (*Grosche et al., 2016*). Briefly, retinae were treated with papain (0.2 mg/ml; Roche Molecular Biochemicals) for 30 min at 37°C in the dark in $Ca^{2+}$- and $Mg^{2+}$-free extracellular solution (140 mM NaCl, 3 mM KCl, 10 mM HEPES, 11 mM glucose, pH 7.4), washed several times in extracellular solution and incubated with DNase I (200 U/ml). Afterwards the tissue was triturated in extracellular solution containing (mM) 135 NaCl, 3 KCl, 2 $CaCl_2$, 1 $MgCl_2$, 1 $Na_2HPO_4$, 10 HEPES, and 11 glucose adjusted to pH 7.4 with Tris. After centrifugation, cells were resuspended and incubated in extracellular solution containing biotinylated hamster anti-CD29 (clone Ha2/5, 0.1 mg/ml, BD Biosciences, Heidelberg, Germany) for 15 min at 4°C. Cells were washed in extracellular solution, spun down, resuspended in the presence of anti-biotin MicroBeads (1:5; Miltenyi Biotec, Bergisch Gladbach, Germany) and incubated for 10 min at 4°C. After washing, CD29 +Müller cells were separated using large cell (LS) columns according to the manufacturer's instructions (Miltenyi Biotec). To purify microglial and vascular cells in addition to Müller cells, the retinal cell suspension was subsequently incubated with CD11b- and CD31 microbeads (Miltenyi Biotec) and depleted from the retinal suspension using LS-columns prior to Müller cell enrichment.

## qRT-PCR

Total RNA was isolated from whole brain tissue using the PureLink RNA Micro Scale Kit (Thermo Fisher Scientific, Germany). Upon DNase-treatment to remove genomic DNA, first-strand cDNAs were synthesized from 50 ng of total RNA (RevertAid H Minus First-Strand cDNA Synthesis Kit, Fermentas by Thermo Fisher Scientific, Germany). Primers (see *Supplementary file 2*) were designed using the Universal Probe Library Assay Design Center (Roche). Transcript levels of candidate genes were measured by qRT-PCR using the QuantStudio 5 Real-Time PCR system (384 well, Life Technologies) according to the company's guidelines. All data are expressed as mean ± standard error (SEM) unless stated otherwise. Statistical analyses were performed using Graphpad Prism 7 Software (San Diego, CA). Unless stated otherwise, the significance was determined by the non-parametric Mann-Whitney U test.

## Retinal ischemia

Transient retinal ischemia was induced in one eye by the HIOP (high intraocular pressure) method as previously described (*Pannicke et al., 2014*). The other eye remained untreated as internal control. Anesthesia was induced with ketamine (100 mg/kg body weight, intraperitoneal (ip); Ratiopharm, Ulm, Germany), xylazine (5 mg/kg, ip; Bayer Vital, Leverkusen, Germany), and atropine sulfate (100 mg/kg, ip; Braun, Melsungen, Germany). The anterior chamber of the test eye was cannulated from the pars plana with a 30-gauge infusion needle, connected to a saline bottle. The intraocular pressure was increased to 160 mmHg for 60 min by elevating the bottle. After removing the needle, the

animals survived for 3 days and, subsequently, were sacrificed with carbon dioxide. Mice were in the C57BL/6N background.

## Stab wound injury

Stab wound injury was performed in the somatosensory cortex, as previously described (*Heinrich et al., 2014*; *Heimann et al., 2017*). Briefly, anesthetized animals received a stab wound of the somatosensory cortical GM with a lancet-shaped knife (Alcon) Coordinates from bregma: AP −0.8 to −2.0, ML 1.6 to 2.0 mm and DV −0.6. Animals were allocated to experimental groups regarding their genotype and kept under standard conditions with access to water and food *ad libitum*. Five days post injury (dpi), animals were transcardially perfused and brains processed for immunohistochemistry as described above.

For analysis, seven corresponding slices were prepared from each animal and triple staining of GFAP, NeuN, and Iba1 were performed sequentially (starting with NeuN and over night fixation, and followed by GFAP and Iba1 labeling). Confocal images were taken at identical exposure settings with single channel maximum intensity projections set to automatic threshold. Iba1- Neu-, and DAPI-positive areas were measured using NIH ImageJ software (Image > adjust > threshold; Analyse > measure). Iba1-positive areas and NeuN-negative/lesioned areas were normalized to DAPI-positive areas. Data were analyzed using Graphpad Prism 7.

## Intra-amygdala kainic acid-induced status epilepticus mouse model

Procedures were undertaken as described previously (*Jimenez-Pacheco et al., 2013*) in 8–12 week-old mice (line 17/FVB/N) bred at the Biomedical Research Facility at RCSI. Mice were anesthetized with isoflurane (5% induction, 1–2% maintenance) and maintained normothermic by means of a feedback-controlled heat blanket (Harvard Apparatus Ltd, Kent, UK). Fully anesthetized, mice were placed in a stereotaxic frame and a midline scalp incision was performed to expose the skull. A guide cannula (coordinates from Bregma; AP = −0.94 mm, L = −2.85 mm) for intra-amygdala kainic acid (Sigma Aldrich, Dublin, Ireland) injection was fixed in place with dental cement and status epilepticus induced in fully awake mice via microinjection of 0.3 µg KA (in 0.2 µl phosphate-buffered saline) into the basolateral amygdala. Control animals received 0.2 µl PBS. 40 min after injections, the anticonvulsive lorazepam (6 mg/kg, Wyetch, Taplow, UK) was delivered i.p. to curtail seizures and reduce morbidity and mortality. Mice were killed 24 hr after lorazepam injection and perfused (4% PFA in PBS). Brains were post-fixed overnight in 4% PFA, embedded in 2% agarose, and cut by vibratome in 30 µm sections. Sections were stored in glycol at −20° C until use.

## Behavioral experiments

Experiments were performed with 10–13 weeks old mice on 3 consecutive days in the following order.

### Balance beam

Mice were positioned in the middle of a 50 cm long and round (1 cm diameter) wooden bar, which was fixed 44 cm above a padded surface between two 14 × 10 cm wooden escape platforms. The time for which animals stayed on the beam was measured and if a platform was reached, 60 s were counted. Each test was performed three times for a duration of 60 s.

### String suspension

A 3-mm-thick rope was loosely attached to the balance beam platforms in 35 cm height. Mice were hold in front of the middle of the string so that they could grab it with their fore-paws. The following scoring was used during a 60 s test duration: 0 = unable to stay on rope; 1 = hanging on rope with one pair of paws; 2 = like 1, but with attempt to climb; 3 = sitting on rope, keeping balance; 4 = rope grabbed with all paws and tail together with a laterally movement of the mouse; 5 = escape on platform.

### Hot plate

The plate was set to 50°C and surface temperature continuously monitored with a digital thermometer. Time measurement started when the animal was gently placed on the hot plate and stopped

when the animal licked its hind paw for the first time or jumped off. Animals that failed to react within 60 s were removed from the plate to avoid thermal injury, and were assigned the value of 60 s. Each test was performed three times with at least 15 min intervals between measurements.

## Animals

Mice were housed in standard conditions (22°C, 12 hr light–dark cycle, water/food *ad libitum*). Animal handling and experimental procedures were performed in accordance with German and European Union guidelines and were approved by the State of Upper Bavaria (stab wound injury (55.2.1.54-2532-171-11), retinal ischemia (TVV 54/12; 55.2 DMS-2532-2-182), transcardial perfusion (55.2-1-54-2532-59-2016)) and Lower Saxony (generation of BAC transgenic mice, transcardial perfusion (33.9-42502-04-12/0863), behavioral experiments (3392 42502-04-13/1123)). Status epilepticus was induced in accordance with the principles of the European Communities Council Directive (86/609/EEC) and procedures reviewed and approved by the Research Ethics Committee of the Royal College of Surgeons in Ireland (REC 205 and 1322) and performed under license from the Department of Health and Children, Ireland. All efforts were made to minimize suffering and number of animals used.

## Acknowledgements

We thank Conny Neblung, Sarah Schlagowski, Annett Sporning, Irina Zamolo, and Heinz Janser for technical assistance, Heinrich Betz and Walter Stühmer for generously providing support and facilities, Stephan Kröger for help with NMJ preparation, Ursula Fünfschilling for advice in animal breeding, Ablynx for permission to use mouse P2×7 specific nanobody 7E2, Klaus Armin Nave for FLIR, EIIa-Cre, and cnp1-Cre mice, and Steffen Jung for Cx3cr1-Cre mice. We are also grateful to Susanne Koch, Nundehui Diaz Lezama, and Ali Rifat for their commitment during the revision. This work was financed by the DFG (Ni 592/4–5 and 7 (AN), No 310/11–1 (FK-N), and SFB 1328 (AN, FK-N, TM), the European Union's Horizon 2020 research and innovation programme under the Marie Sklodowska-Curie grant agreement (No 766124 (TE, AN), and the Science Foundation Ireland (13/SIRG/2098 and 17/CDA/4708 (TE)). SB was granted a Faculty Research Visit Grant from the DAAD (German Academic Exchange Service).

## Additional information

### Competing interests

Friedrich Koch-Nolte: FKN receives a share of antibody sales via MediGate GmbH, a wholly owned subsidary of the University Medical Center Hamburg-Eppendorf. FKN is a co-inventor on patent applications on P2X7-specific nanobodies (WO2010070145, WO2013178783). The other authors declare that no competing interests exist.

### Funding

| Funder | Grant reference number | Author |
| --- | --- | --- |
| Deutsche Forschungsgemeinschaft | Ni 592/4-5 | Annette Nicke |
| Deutscher Akademischer Austauschdienst | | Santina Bruzzone |
| Deutsche Forschungsgemeinschaft | No 310/11-1 | Friedrich Koch-Nolte |
| Science Foundation Ireland | 13/SIRG/2098 | Tobias Engel |
| Horizon 2020 Framework Programme | 766124 | Tobias Engel Annette Nicke |
| Deutsche Forschungsgemeinschaft | SFB 1328 | Tim Magnus Friedrich Koch-Nolte Annette Nicke |
| Science Foundation Ireland | 17/CDA/4708 | Tobias Engel |

| Deutsche Forschungsge- | Ni 592/4-7 | Annette Nicke |
| meinschaft | | |

The funders had no role in study design, data collection and interpretation, or the decision to submit the work for publication.

## Author contributions

Karina Kaczmarek-Hajek, Conceptualization, Formal analysis, Investigation, Visualization, Methodology, Writing—review and editing; Jiong Zhang, Formal analysis, Validation, Investigation, Visualization, Methodology, Writing—review and editing; Robin Kopp, Formal analysis, Validation, Investigation, Visualization, Writing—review and editing; Antje Grosche, Formal analysis, Investigation, Visualization, Methodology, Writing—review and editing; Björn Rissiek, Santina Bruzzone, Swetlana Sirko, Formal analysis, Investigation, Methodology, Writing—review and editing; Anika Saul, Investigation, Visualization, Writing—review and editing; Tobias Engel, Christine Stadelmann, Friedrich Koch-Nolte, Resources, Methodology, Writing—review and editing; Tina Jooss, Investigation, Visualization; Anna Krautloher, Validation, Investigation, Writing—review and editing; Stefanie Schuster, Formal analysis, Investigation; Tim Magnus, Resources, Funding acquisition; Volker Eulenburg, Conceptualization, Formal analysis, Investigation, Methodology, Writing—review and editing; Annette Nicke, Conceptualization, Resources, Formal analysis, Supervision, Funding acquisition, Investigation, Visualization, Methodology, Writing—original draft, Project administration

## Author ORCIDs

Robin Kopp http://orcid.org/0000-0002-1639-2868
Björn Rissiek http://orcid.org/0000-0001-5327-5479
Swetlana Sirko http://orcid.org/0000-0001-5950-616X
Volker Eulenburg http://orcid.org/0000-0002-4878-5746
Annette Nicke http://orcid.org/0000-0001-6798-505X

## Ethics

Animal experimentation: Animal handling and experimental procedures were performed in accordance with German and European Union guidelines and were approved by the State of Upper Bavaria (stab wound injury (55.2.1.54-2532-171-11), retinal ischemia (TVV 54/12; 55.2 DMS-2532-2-182), transcardial perfusion (55.2-1-54-2532-59-2016)) and Lower Saxony (generation of BAC transgenic mice, transcardial perfusion (33.9-42502-04-12/0863), behavioral experiments (3392 42502-04-13/1123)). Status epilepticus was induced in accordance with the principles of the European Communities Council Directive (86/609/EEC) and procedures reviewed and approved by the Research Ethics Committee of the Royal College of Surgeons in Ireland (REC 205 and 1322) and performed under license from the Department of Health and Children, Ireland. All efforts were made to minimize suffering and number of animals used.

## Decision letter and Author response

Decision letter https://doi.org/10.7554/eLife.36217.029
Author response https://doi.org/10.7554/eLife.36217.030

# Additional files

## Supplementary files

• Supplementary file 1. List of probes.
DOI: https://doi.org/10.7554/eLife.36217.025

• Supplementary file 2. List of primers.
DOI: https://doi.org/10.7554/eLife.36217.026

• Transparent reporting form
DOI: https://doi.org/10.7554/eLife.36217.027

## Data availability

All data generated or analysed during this study are included in the manuscript and supporting files.

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
