## [Decision Letter]

Thank you for submitting your article "Re-evaluation of neuronal P2X7 expression using novel mouse models and a P2X7-specific nanobody" for consideration by *eLife*. Your article has been reviewed by three peer reviewers, and the evaluation has been overseen by Kenton Swartz as the Reviewing Editor and Richard Aldrich as the Senior Editor. The following individual involved in review of your submission has agreed to reveal their identity: Thomas Grutter (Reviewer #2).

The reviewers have discussed the reviews with one another and the Reviewing Editor has drafted this decision to help you prepare a revised submission.

Summary:

There are many incongruous reports on P2X7 expression patterns that have engendered controversy over how P2X7 deletion/inhibition impacts disease. This murkiness limits the clinical advancement of P2X7 inhibitors and obscures our mechanistic understanding. This manuscript by Kaczmarek-Hájek et al. is a rigorous examination of P2X7 protein expression throughout the nervous system. The authors have generated P2X7-EGFP fusion BAC mice, demonstrating that the fusion protein is functional and trafficked appropriately. The authors' findings regarding which cell types express P2X7 are in conflict with several reports in the literature. Much of the discrepancy can be explained by the overreliance of past works on non-specific commercial antibodies, the weaknesses of which are exposed here with proper controls (Figure 3—figure supplement 6, Figure 2E). Several transgenic mouse lines have previously been used to monitor *P2rx7* expression, but show limited agreement with each other or the present report. There are several reasons to accept the present work as the authoritative reference for P2X7 expression patterns. First, the GFP-fusion BAC transgenic informs protein expression, which is an improvement upon previous transgenics and RNA-seq datasets. Second, BAC fusion reporter lines represent arguably the most sensitive modern technique for describing protein expression that still maintains high signal specificity. Third, the present analysis is well controlled, often corroborating findings with a validated nanobody or multiple transgenic lines. Finally, this work provides a broad characterization across many regions of the nervous system, avoiding overgeneralization of findings from a focal investigation. Overall, the experiments are very well performed, and data very well analyzed. The results are clearly presented and the paper is well written and accessible to the general reader. The authors should be congratulated for this huge amount of work.

We have some concerns (enumerated below) about the authors' limited investigation of region-specific P2X7 expression patterns and unsatisfying characterization of injury-induced changes. However, if these issues can be addressed, we would support publication of this work in *eLife* as a well-executed contribution that will be informative to a broad range of disciplines concerned with P2X7 function.

Essential revisions:

1) Although the authors provide evidence that the EGFP-tagged P2X7 construct is functional by a dye uptake assay, they should also show ATP-induced currents by electrophysiology. These data would help the reader to appreciate the influence of the EGFP-tag on activation and deactivation kinetics of the chimeric channel.

2) The authors observe a band of GFP signal in CA3/dentate gyrus (e.g. bottom-right panel in Figure 3—figure supplement 1B) that would appear to be consistent with functional expression of P2X7 in a subset of neurons as reported (Engel et al., 2012 and Jimenez-Pacheco, 2013). Lack of neuronal expression is a primary claim, so it is important to know: what cell type or structure can this staining be attributed to? We would be satisfied if the authors showed synaptic marker co-staining with the CA3/DG GFP signal, or perhaps include a discussion about that signal that rules out neuronal contribution.

3) There is strong GFP signal in the adult basal ganglia/hypothalamus and ventral pons that do not seem to be explained by microglia/OPCs (Figure 2A, Figure 2—figure supplement 1). Protein in these regions should also be characterized if the present work is to be considered a comprehensive analysis of P2X7 expression. We would be satisfied with NeuN-GFP co-staining in basal ganglia, hypothalamus, and pons along with TH-GFP co-staining in the basal ganglia.

4) The authors image apparently non-glial P2X7 in other CNS regions, but don't uncover the source. What cell type is responsible for the S100β- P2X7+ signal in cerebellum?

5) The authors do not look for changes in which cell types produce P2X7 protein in the retinal injury model and analysis of changes in GFP expression after the stab injury are obstructed by autofluorescence induced by the injury. FACS or chromogenic staining should be used to clarify this, especially since neuronal expression was only detected after kainite-induced seizures. Alternatively, the authors should comment on the potential obfuscation of GFP signal due to injury-induced autofluorescence in Figure 5C.

6) The authors make statements about trends in altered inflammatory responses in the context of P2X7 overexpression (text related to Figure 5), but their N is too small to justify meaningful claims. The low n makes the conclusions about post-injury impact of P2X7 ambiguous, but this is not essential to the work's major conclusions and this concern could be addressed with changes to the discussion of these data.

7) The number of animals used throughout Figure 5 is insufficient to support substantive claims. Also, several supplementary stainings are conducted with n=3-6 sections from n=1 animal; these stainings are important and should be repeated in tissue from at least 2 separate animals.

---

## [Author Response]

Essential revisions:1) Although the authors provide evidence that the EGFP-tagged P2X7 construct is functional by a dye uptake assay, they should also show ATP-induced currents by electrophysiology. These data would help the reader to appreciate the influence of the EGFP-tag on activation and deactivation kinetics of the chimeric channel.

A figure showing ATP-induced currents from patch clamp recordings of P2X7 (wt) and P2X7-EGFP expressing HEK cells has been introduced as Figure 1E in the Figure 1—figure supplement 1.

2) The authors observe a band of GFP signal in CA3/dentate gyrus (e.g. bottom-right panel in Figure 3—figure supplement 1B) that would appear to be consistent with functional expression of P2X7 in a subset of neurons as reported (Engel et al., 2012 and Jimenez-Pacheco, 2013). Lack of neuronal expression is a primary claim, so it is important to know: What cell type or structure can this staining be attributed to? We would be satisfied if the authors showed synaptic marker co-staining with the CA3/DG GFP signal, or perhaps include a discussion about that signal that rules out neuronal contribution.

In the references mentioned above, P2X7 expression was described in individual hippocampal (CA1 and DG) and cortical neurons of the granule layer whereas Metzger, 2016 describes extensive RNA expression in the granule layer of the CA3 region. In our P2X7-EGFP BAC transgenic mice, a clear band of GFP signal is regularly detected in the molecular layer of the dentate gyrus and, to a lesser extent in the CA3 region (e.g. Figure 2A, Figure 3—figure supplement 1A (now B) bottom-right panel). To further exclude neuronal expression of P2X7-EGFP in the CA3 and DG regions, we added, as suggested, co-stainings with synaptic markers of these regions (Figure 3—figure supplement 1 and 2). Based on our data, we attribute the more intense stainings to a higher density and/or more ramified morphology of microglia in these regions. In particular, they seem to accumulate directly at the border to the granular layer in the DG (Figure 3—figure supplement 3A). In support of our explanation and in contrast to the much larger area seen in the GenSat mouse (e.g. http://www.gensat.org/imagenavigator.jsp?imageID=29030) the thickness of the band in this region equals the radius of microglia with their extensions. The text has been changed accordingly (subsection “Analysis of P2X7-EGFP localization in the brain”).

3) There is strong GFP signal in the adult basal ganglia/hypothalamus and ventral pons that do not seem to be explained by microglia/OPCs (Figure 2A, Figure 2—figure supplement 1). Protein in these regions should also be characterized if the present work is to be considered a comprehensive analysis of P2X7 expression. We would be satisfied with NeuN-GFP co-staining in basal ganglia, hypothalamus, and pons along with TH-GFP co-staining in the basal ganglia.

As suggested, co-stainings of P2X7-EGFP with antibodies against NeuN and TH were performed. These are now included as Figure 3—figure supplement 5 and referred to in subsection “Analysis of P2X7-EGFP localization in the brain”. No co-localization of GFP and neuronal markers were found, but the morphology of the labeled cells are consistent with oligodendroglial or microglial expression of EGFP.

4) The authors image apparently non-glial P2X7 in other CNS regions, but don't uncover the source. What cell type is responsible for the S100β- P2X7+ signal in cerebellum?

The fact that we see EGFP immunoreactivity that does not co-localize with S100β immunoreactivity does not necessarily mean that the EGFP is not of glial origin. To identify the responsible cell types, we performed co-staining with calbindin and vGlut2 (Purkinje cells and their glutamatergic synapses) as well as GFAP and S100β (Bergmann glia). The latter are now included in Figure 2—figure supplement 2C.

However, due to a very dense and uniform EGFP signal and a higher autofluorescence background, co-localization of the EGFP-signal in adult cerebellum with these markers was not possible. Based on the co-localization of EGFP with S100β-positive cells in P7 animals, in which the EGFP signal is less diffuse and more structured, we conclude that P2X7-EGFP is expressed in Bergmann glia and that the formation of Bergmann glia microdomains after the first week of life (Grosche et al., 2002) results in the observed diffuse and dense staining pattern.

In agreement with our conclusion, co-staining of acutely dissociated cerebellar cells from adult animals with Iba1 and S100β confirmed the presence of EGFP in S100β and Iba1-positive cells (i.e. Bergmann glia and Microglia, respectively). These data are now included in Figure 2—figure supplement 2. The text has been changed accordingly in subsection “Analysis of P2X7-EGFP localization in the brain”.

5) The authors do not look for changes in which cell types produce P2X7 protein in the retinal injury model and analysis of changes in GFP expression after the stab injury are obstructed by autofluorescence induced by the injury. FACS or chromogenic staining should be used to clarify this, especially since neuronal expression was only detected after kainite-induced seizures. Alternatively, the authors should comment on the potential obfuscation of GFP signal due to injury-induced autofluorescence in Figure 5C.

a) We have now included co-stainings for EGFP with glutamine synthetase and Iba1 in the ganglion cell layer to exclude Müller cell and neuronal expression and confirm restriction to microglia in the ischemic retina model. In addition, we included cell-type specific quantitative PCR data from treated and non-treated retinae of wt and transgenic animals to exclude upregulation of P2X7 in neurons and other retinal cell types. These data are included in Figure 5—figure supplement 1 A-C and referred to in the text (subsection “Consequences of P2X7 overexpression under physiological and pathological conditions”). The original Figure 5A right panel (P2X7 expression in microglia) was included into the new Figure 5—figure supplement 1C to demonstrate the prominent P2X7 expression in microglia in direct comparison to other retinal cell types.

b) Due to autofluorescence in the area that is immediately adjacent to lesion core and filled with inflammatory cells (~ 0-75μm) we cannot exclude neuronal P2X7 expression. However, in the penumbra that immediately interfaces with, and surrounds, damaged and inflamed tissue lesions after injury (>100μm) no upregulation in NeuN+ neurons was found. Unfortunately, no additional animal experiments, and hence, no chromogenic stainings of injured tissues could be performed in the given time. Therefore, we commented on the potential obfuscation (subsection “Consequences of P2X7 overexpression under physiological and pathological conditions” and legend of Figure 5D), as suggested.

c) In addition, we obtained slices from mice after kainic acid-induced seizures, which confirmed the lack of P2X7 protein upregulation in neurons in the dentate gyrus, CA1, and CA3 regions. These are now included in Figure 5—figure supplement 2 and referred to in subsection “Consequences of P2X7 overexpression under physiological and pathological conditions”.

6) The authors make statements about trends in altered inflammatory responses in the context of P2X7 overexpression (text related to Figure 5), but their N is too small to justify meaningful claims. The low n makes the conclusions about post-injury impact of P2X7 ambiguous, but this is not essential to the work's major conclusions and this concern could be addressed with changes to the discussion of these data.

The data in Figure 5 are now discussed more carefully in the text related to Figure 5 in subsection “Consequences of P2X7 overexpression under physiological and pathological conditions”. In particular, the limited number of animals used in these preliminary experiments and the need for confirmation are now pointed out in the text.

7) The number of animals used throughout Figure 5 is insufficient to support substantive claims. Also, several supplementary stainings are conducted with n=3-6 sections from n=1 animal; these stainings are important and should be repeated in tissue from at least 2 separate animals.

Unfortunately, additional animal experiments could not be performed in the given time and suitable in vitro models could not successfully be established due to a limited number of available animals. However, we repeated all stainings on additional animals in cases were less than three animals had been used before.

In cases were clearer stainings were obtained, these were used to replace the original ones in the respective figures (Figure 3—figure supplement 1 and 2).